

# Deriving seismic velocities on the micro-scale from c-axis orientations in ice cores

Johanna Kerch[1,2], Anja Diez[3], Ilka Weikusat[1,4], and Olaf Eisen[1,5]

[1]Alfred Wegener Institute Helmholtz Centre for Polar and Marine Research, 27568 Bremerhaven, Germany
[2]Institute of Environmental Physics, Heidelberg University, 69120 Heidelberg, Germany
[3]Norwegian Polar Institute, Tromsø, Norway
[4]Department of Geosciences, Eberhard-Karls-University Tübingen, 72074 Tübingen, Germany
[5]Fachbereich Geowissenschaften, Universität Bremen, Bremen, Germany

*Correspondence to:* Johanna Kerch (jkerch@awi.de)

**Abstract.** One of the greatest challenges in glaciology, with respect to sea level predictions, is the ability to gain information on bulk ice anisotropy in ice sheets and glaciers, which is urgently needed to improve our understanding of ice-sheet dynamics. Therefore, we investigate the effect of crystal anisotropy on seismic velocities in a glacier. We revisit the framework which is based on fabric eigenvalues to derive approximate seismic velocities by exploiting the assumed symmetry. In contrast to previous studies, we calculate the seismic velocities using the exact c-axis angles describing the orientations of the crystal ensemble in an ice-core sample. We apply this approach to fabric data sets from an Alpine (KCC) and a polar (EDML) ice core. The results allow a quantitative evaluation of the earlier approximative eigenvalue framework. Additionally, our findings highlight the variation in seismic velocity as a function of the horizontal azimuth of the seismic plane, which can be significant in case of non-symmetric orientation distributions and results in a strong azimuth-dependent shear-wave splitting. For the first time, we assess the change in seismic anisotropy that can be expected on a short spatial scale in a glacier due to a strong variability in crystal-orientation fabric. Our investigation of seismic anisotropy based on ice-core data contributes to advancing the interpretation of seismic data, with respect to extracting bulk information about crystal anisotropy without having to drill an ice core and with special regard to future applications employing ultrasonic sounding.

## 1 Introduction

One of the most important goals for glaciological research is the establishment of a thorough understanding of the ice dynamics for which the internal deformation plays a crucial role. This deformation is predominantly evident and described on a macro-scale ($\sim$ km). However, it is necessary to connect the bulk behaviour with the governing processes on the micro-scale ($\sim$ μm) to be able to develop a comprehensive understanding of the deformation mechanisms (Weikusat et al., 2017). The fundamental deformation mechanisms on the atomic scale are driven by the external stress field and lead to the alignment of single ice crystals in preferential directions (Faria et al., 2014). Due to the intrinsic anisotropy of each ice crystal an anisotropic bulk medium is formed as a result of the crystal-preferred orientation (CPO, also known as lattice-preferred orientation, LPO, and crystal-orientation fabric, COF). The anisotropy is evident in elastic, plastic and electromagnetic properties of the ice and the





respective parameters can be connected to each other. The plastic anisotropy can have a considerable influence on the bulk deformation rate and vice versa. Hence, it is desirable to incorporate the development of anisotropy in flow models (Pettit et al., 2007; Seddik et al., 2008; Martin et al., 2009).

Currently, the development and extent of fabric anisotropy is mainly investigated by laboratory measurements on ice core
samples which provide one-dimensional data (along the core axis) that only partially cover the length of the core. However, geophysical evidence of crystal-orientation fabric can also be obtained by exploiting the elastic anisotropy which influences the propagation of seismic waves in ice (Blankenship and Bentley, 1987; Smith et al., 2017). Seismic reflections occur due to sudden changes of fabric (Horgan et al., 2008, 2011; Hofstede et al., 2013) and offer the chance to obtain information on the COF structure in various depths of the ice column and laterally extended, towards a full 3d information of anisotropy in ice
sheets and glaciers, which will never be feasible via the drilling of an ice core.

The motivation of this study is to improve the interpretation of seismic data by connecting the micro- and the macro-scale using the elastic properties of ice. Early work to this end were accomplished by Bennett (1968); Bentley (1972); Blankenship and Bentley (1987) and more recent approaches include Gusmeroli et al. (2012) and Diez and Eisen (2015). Specifically, the starting point of this paper is the study by Diez and Eisen (2015), who establish a connection between the commonly used
fabric parameter of second-order orientation tensor eigenvalues and the elasticity tensor describing the polycrystalline medium to calculate seismic velocities from ice-core fabric data (*ev* framework). They illustrate the proposed calculation framework on fabric data from the polar ice core EDML (from the drilling EPICA in Dronning Maud Land, Antarctica). The main objective of the here presented study is to provide a refined algorithm for the estimation of the systematic deviation made by using fabric eigenvalues for the calculation of the elasticity tensor and the derived seismic velocities.

We first present experimental measurements, theoretical basis and mathematical algorithm of our new framework (*cx*). We apply this framework to two ice cores and further investigate how fabric variability on the submetre scale is reflected in theoretical seismic interval velocities. Finally, we assess the effect of asymmetrical fabric distributions and explore the potential of azimuth-dependent seismic surveys.

## 2   Methodology

### 2.1   Laboratory ice fabric measurements

For our analysis of seismic velocities we use fabric data from the polar ice core EDML and the Alpine ice core KCC. The EDML ice core was drilled as part of EPICA (European Project for Ice Coring in Antarctica) until 2006 at Kohnen Station, Antarctica, and reaches to a depth of 2774 m (Oerter et al., 2009; Weikusat et al., 2017). The KCC ice core was drilled in 2013 on the Alpine glacier Colle Gnifetti, Monte Rosa Massif, Switzerland/Italy (N 45°55.736, E 7°52.576, 4484 m a.s.l.) in
about 100 m distance to the ice core KCI, drilled in 2005 (Bohleber et al., 2017, in press). KCC is 72 m long with the firn-ice-transition at a depth of about 36 m and a borehole temperature between −13.6 °C in 13 m depth and −12.4 °C at the bed rock measured in 2014 (pers. comm. M. Hoelzle, University of Fribourg, 2014). Both KCC and EDML were stored at minimum −18 °C during transport and at −30 °C during processing.



Vertical and horizontal thin sections of the ice cores were prepared and measured by means of polarised light microscopy (e.g. Wilson et al., 2003; Peternell et al., 2009) with an automatic fabric analyser from Russell-Head Instruments (models G20 and G50 in case of EDML and G50 for KCC). For each identified ice crystal in the thin section the measurement provides the orientation of the crystallographic c-axis by two angles, azimuth $\vartheta$ in the interval $(0, 2\pi)$ and colatitude $\varphi$ in the interval

$(0, \pi/2)$, with respect to the vertical ice-core axis that we define to coincide with the z-axis of the global coordinate system (Fig. 1). The c-axis is expressed as a vector in spherical coordinates with unit length:

$$\boldsymbol{c}(\vartheta, \varphi) = (\sin(\varphi)\cos(\vartheta), \sin(\varphi)\sin(\vartheta), \cos(\varphi)) \tag{1}$$

The EDML fabric data (data sets: Weikusat et al., 2013a, b, c, d) is presented in detail in Weikusat et al. (2017). The total data set used in this study comprises 154 samples between 104 and 2563 m depth with a coarse sampling interval and 40 additional

vertical section samples that were measured continuously in several intervals between 2359 and 2380 m. These high resolution measurements were done with the G50 instrument. The KCC fabric data (data set: Kerch et al., 2016) consists of 85 vertical thin sections and covers 11 % of the entire ice core.

Eigenvalues $\lambda_i$ ($i = 1, 2, 3$) of the second-order orientation tensor $a_{ij}^{(2)}$ are usually calculated from the c-axis distribution within a thin section sample and can be grain-, area- or volume-weighted to describe the fabric (Woodcock, 1977; Durand et al., 2006;

Mainprice et al., 2011). They describe the type and strength of anisotropy in the crystal ensemble visible in the thin section (e.g. Cuffey and Paterson, 2010). Typically, different types of fabric are identified (Diez and Eisen, 2015) by the relation of the three eigenvalues with $\lambda_1 \leq \lambda_2 \leq \lambda_3$ and $\sum \lambda_i = 1$. By this classification the crystal anisotropy of the bulk can be described in a convenient way, if a unimodal distribution can be assumed, and can be associated with different deformation regimes (e.g. Weikusat et al., 2017).

## 2.2   Seismic wave propagation in anisotropic ice

In a glacier, the fabric anisotropy also introduces an anisotropy of the elastic properties of the material. This elastic anisotropy results in a seismic anisotropy, which means the propagation of seismic waves is influenced by the fabric anisotropy. To study this connection, theoretical velocities can be calculated if the fabric anisotropy is known.

The mathematical background for the calculation of seismic phase velocities from the elastic properties in anisotropic ice can

be found in many publications (e.g. Tsvankin, 2001; Diez and Eisen, 2015). For convenience the essential concepts are shortly repeated in the following. Group velocities are not subject of this study and hence disregarded.

For an anisotropic elastic medium – ice behaves elastically for the propagation of seismic waves – stress and strain are linearly connected following the generalised Hooke's law:

$$\sigma_{mn} = c_{mnop}\tau_{op} \qquad \text{with} \quad m, n, o, p = 1, 2, 3$$

where $c_{mnop}$ is the elasticity tensor, a fourth-order tensor which describes the medium's elastic properties. The inverse relation uses the compliance tensor $s_{mnop}$. Due to the symmetry of strain and stress tensor and thermodynamic considerations (Aki and Richards, 2002), the 81 unknowns of the elasticity tensor reduce to 21 independent components for general anisotropy.



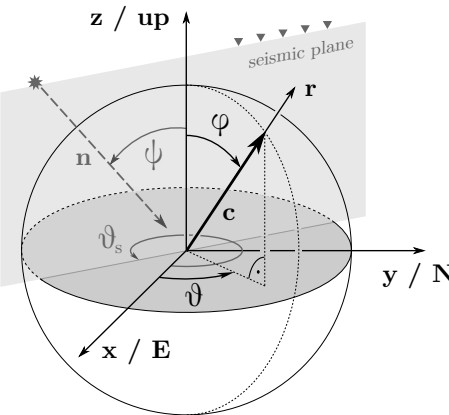

**Figure 1.** The global coordinate system $\{x, y, z\}$ used for the description of a c-axis with unit length: spherical coordinates $\vartheta$ and $\varphi$ specify the orientation of the c-axis. For each grain the c-axis can be expressed in its local coordinate system $\{p, q, r\}$ by $(0, 0, 1)$; $p, q$ are not shown here. The equatorial plane (dark grey) corresponds to the horizontal thin section plane. The $z$-axis is assumed to be parallel to the ice core axis. The orientation of a hypothetical seismic plane (light grey) is defined by the seismic azimuth angle $\vartheta_\mathrm{s}$, with the angle $\psi$ of an incident seismic wave (dashed line).

The elasticity tensor can then be expressed in a simplified manner, known as Voigt notation (Voigt, 1910), where pairs of indices from the fourth-order tensor are replaced by single indices. The resulting elasticity tensor in Voigt notation $C_{ij}$ ($i, j = 1, 2, 3, 4, 5, 6$) is a symmetric second-order tensor. In case of monocrystalline ice I$_\mathrm{h}$, the components of the elasticity tensor were measured in the laboratory. There are five independent components due to the hexagonal crystal symmetry. Several

5    sets of values for the elastic moduli have been found by different authors, as summarised in Diez et al. (2015). Here, the monocrystal elasticity tensor $\mathbf{C}_\mathrm{m}$ by Gammon et al. (1983), as measured on samples of artificial ice at $-16\,^\circ\mathrm{C}$ by means of Brillouin spectroscopy, is used for all calculations:

$$\mathbf{C}_\mathrm{m} = \begin{bmatrix} 13.929 & 7.082 & 5.765 & 0 & 0 & 0 \\ 7.082 & 13.929 & 5.765 & 0 & 0 & 0 \\ 5.765 & 5.765 & 15.010 & 0 & 0 & 0 \\ 0 & 0 & 0 & 3.014 & 0 & 0 \\ 0 & 0 & 0 & 0 & 3.014 & 0 \\ 0 & 0 & 0 & 0 & 0 & 3.424 \end{bmatrix} \cdot 10^9\,\mathrm{N/m^2} \qquad (2)$$

To apply this description to the study of large ice sheets and glaciers, we have to consider the elastic properties of the polycrys-

10    tal. The understanding of the elastic behaviour of a monocrystal can be used together with the fabric description to estimate the elastic properties of the polycrystal. Different theoretical models have been developed for the estimation of the elasticity tensor of an anisotropic polycrystal, usually making use of fabric symmetries (e.g. Nanthikesan and Sunder, 1994; Maurel et al., 2015) or by calculating orientation density functions (ODF, Mainprice et al., 2011). In this context some authors refer to the polycrystal as "effective medium" (Maurel et al., 2015).





For the calculation of the polycrystal elastic properties from anisotropic monocrystal properties the concept of Voigt-Reuss-bounds is often used. They provide estimates of the upper and lower limits for the elastic moduli of the polycrystal, as was shown by Hill (1952), with the Reuss bound exceeding the Voigt bound. Nanthikesan and Sunder (1994) find that the difference of the Voigt-Reuss-bounds for the elastic moduli of polycrystalline ice does not exceed 4.2 % and conclude that either of the bounds or an average is a good approximation.

Once the elastic properties for the polycrystal are known, the Christoffel equation provides the relationship to calculate seismic velocities. For a linearly elastic, arbitrarily anisotropic homogeneous medium the wave equation is solved by a harmonic steady-state plane wave and we obtain the Christoffel equation:

$$\begin{bmatrix} G_{11} - \rho v_{\mathrm{ph}}^2 & G_{12} & G_{13} \\ G_{21} & G_{22} - \rho v_{\mathrm{ph}}^2 & G_{23} \\ G_{31} & G_{32} & G_{33} - \rho v_{\mathrm{ph}}^2 \end{bmatrix} \begin{bmatrix} U_1 \\ U_2 \\ U_3 \end{bmatrix} = 0 \tag{3a}$$

or $\quad [c_{mnop} n_n n_p - \rho v_{\mathrm{ph}}^2 \delta_{mo}] U_o = 0$ (3b)

with the density $\rho$, the polarisation vector $\boldsymbol{U}$, the phase velocity $v_{\mathrm{ph}}$, the unit vector normal to the plane wavefront $\boldsymbol{n}$, the Kronecker delta $\delta_{mo}$. $G_{mo} = c_{mnop} n_n n_p$ is the positive definite, thus symmetric Christoffel matrix. The vector $\boldsymbol{n}$ indicates the direction of wave propagation and depends on the angle of incidence $\psi$, which is measured from the vertical, and, if applicable, the azimuth angle $\vartheta_s$ between the vertical plane of incidence and the azimuthal orientation of the ice core with respect to the geocoordinates (Fig. 1):

$$\boldsymbol{n} = (\sin(\psi)\cos(\vartheta_s), \sin(\psi)\sin(\vartheta_s), \cos(\psi)) \tag{4}$$

Eq. (3) constitutes an eigenvalue problem for $G_{mo}$. The real and positive eigenvalues are identified with the phase velocities $v_\mathrm{p}, v_{\mathrm{sh}}, v_{\mathrm{sv}}$ for P-wave, SH-wave and SV-wave respectively. Different solutions are proposed, depending on the form of the elasticity tensor. The solution used in this study for an arbitrarily anisotropic medium is outlined in section 2.4.

Instead of interval velocities often the root mean square (RMS) velocity $v_{\mathrm{rms}}$ is considered, which gives the velocity of the homogeneous half-space equivalent to the stack of $N$ horizontal layers (i) to this depth:

$$v_{\mathrm{rms}}(N) = \sqrt{\frac{\sum_{i=1}^{N} [v^{(i)}]^2 t_0^{(i)}}{\sum_{i=1}^{N} t_0^{(i)}}} \tag{5}$$

with the two-way traveltime (TWT) $t_0$ of a seismic wave that travels vertically through a single layer with the corresponding interval velocity $v$. For a layered anisotropic medium a reliable depth-conversion from traveltimes is only feasible if the RMS velocity for zero-offset can be deduced (Diez et al., 2014).

In-situ temperature and density are essential when comparing seismic velocities. However, as this study is focused on the comparison of calculation frameworks that use the same elastic moduli, a temperature correction will generally not be applied.





### 2.3 Recap: Eigenvalue framework

Diez and Eisen (2015) presented a framework for calculating seismic velocities from COF data in form of eigenvalues, which we briefly recapture here. In the following, this framework is referred to as *ev framework* and associated variables are indicated with $^{\text{ev}}$.

#### 2.3.1 From eigenvalues to seismic interval velocities

The *ev* framework can be summarised in three steps:

1. The fabric data in the standard parameterisation of second-order orientation tensor eigenvalues are sorted into three fabric classes (cone, thick girdle, partial girdle), where each is defined by one or two opening angles $\chi, \phi$, symmetrical with respect to the vertical, and enveloping the c-axis distribution of a sample.

2. The opening angles characterising the fabric of each sample are used to integrate the elasticity tensor of a monocrystal, Eq. (2), to obtain the elasticity tensor of the polycrystal, which exhibits an orthorhombic symmetry with respect to the vertical.

3. From the polycrystal elasticity tensor the approximative solutions to the Christoffel equation (3) for the orthorhombic case provided by Daley and Krebes (2004) are applied to obtain seismic interval phase velocities $v_{\text{p}}^{\text{ev}}, v_{\text{sh}}^{\text{ev}}, v_{\text{sv}}^{\text{ev}}$, which can be used for comparison with measured seismic data. Voigt calculation is used following the argument that Reuss and Voigt bounds are close enough.

#### 2.3.2 Uncertainty of *ev* framework

The advantages of this approach are (Diez and Eisen, 2015):

- Eigenvalues are a standard parameter for expressing the strength of fabric and can be directly used for the *ev* framework without additional information about the particular measurement of thin sections from an ice core.

- By restraining to an orthorhombic symmetry the solution to the Christoffel equation can be readily found. No information on the azimuthal orientation of the ice core (relative to any seismic measurements on a glacier) is needed, although this could be considered to improve the results in case of girdle fabric.

However, some uncertainty is inherent in the framework:

- By restraining to an orthorhombic symmetry while using opening angles to describe the c-axis distribution any information on asymmetric fabric (with respect to the vertical) is dismissed and approximation errors are introduced for more asymmetric c-axes distributions.

- In fabric data from ice cores it can be expected that transitions between fabric classes develop mostly gradually, and only in some depths sudden changes occur due to changes in impurity content or deformation regime (Montagnat et al., 2014;



Weikusat et al., 2017). However, the classification into fabric groups based on threshold values for the eigenvalues can introduce artificial discontinuities in the calculated velocity profile.

We will provide a quantitative estimate of the uncertainty of the *ev* framework in the following sections.

## 2.4 C-axes framework

In this study we aim to provide a quantitative estimate of the error introduced by the approximation of the *ev* framework and to assess the potential of the hitherto neglected information for future analyses. For that purpose the exact angle information of each individual c-axis is used in the following to derive the elasticity tensor $\mathbf{C}_\text{p}$ of the polycrystal. Then, the phase velocities in an arbitrarily anisotropic medium are calculated.

### 2.4.1 Calculating the elasticity tensor for discrete crystal ensemble

If not indicated otherwise, elasticity/compliance tensors and velocities are calculated for the effective medium, which, in this study, is typically represented by a thin section comprising a number $N_\text{g}$ of grains of the order of a hundred to a thousand. A data set of COF measurements from an ice core is considered that gives pairs of angles determining the c-axis of each grain $\boldsymbol{c}(\vartheta, \varphi)$ in a grain ensemble per thin section. We apply the following steps to obtain the effective elasticity tensor for a thin section sample:

1. *Transformation of the monocrystal elasticity tensor:* Considering the monocrystal elasticity tensor $\mathbf{C}_{\text{m},k}$, given by Eq. (2), in the $k$-th grain's local coordinate frame $\{p, q, r\}$ with $\boldsymbol{c} = (0, 0, 1)$, a transformation (indicated by $^\text{t}$) to the global coordinate frame $\{x, y, z\}$ by using the angles $\varphi, \vartheta$ is necessary:

$$\mathbf{C}_{\text{m},k}^\text{t} = \mathbf{R}_{\text{C},z}^\top \, \mathbf{R}_{\text{C},y}^\top \, \mathbf{C}_{\text{m},k} \, \mathbf{R}_{\text{C},y} \, \mathbf{R}_{\text{C},z} \tag{6}$$

with rotation matrix $\mathbf{R}_\text{C}$ as given by Eq. (A3) and $\mathbf{R}_\text{C}^\top$ its transpose matrix. $\mathbf{C}_{\text{m},k}^\text{t}$ is unlikely to have vertical transversely

isotropic (VTI) symmetry, as most c-axes in a real fabric do not coincide with the $z$-axis, but will lie obliquely in the $\{x, y, z\}$ coordinate frame.

   2. *Grain area weighting:* If grain size information is available, each transformed monocrystal elasticity tensor $\mathbf{C}_{\text{m},k}^\text{t}$ is multiplied by the grain cross-section area ($A_k$) fraction $f_k = A_k / \sum_k A_k$. Otherwise, it is multiplied by $1/N_\text{g}$ for an equal contribution of all grains to the effective medium elasticity tensor.

3. *Discrete summation over the transformed monocrystal elasticity tensor for all grains to obtain the polycrystal elasticity tensor $\mathbf{C}_\text{p}$ in the global coordinate frame:*

$$\mathbf{C}_\text{p} = \sum_k \mathbf{C}_{\text{m},k}^\text{t} \tag{7}$$

The obtained elasticity tensor $\mathbf{C}_\text{p}$ is very likely to have only non-zero components and describes an arbitrarily anisotropic medium.

*Derivation via the compliance tensor:* For the aim of considering Reuss and Voigt bounds as introduced above, the polycrystal elasticity tensor is also calculated via the compliance tensor $\mathbf{S}_{\mathrm{m}}$, i.e. the monocrystal elasticity tensor is inverted: $\mathbf{S}_{\mathrm{m}} = \mathbf{C}_{\mathrm{m}}^{-1}$. Steps 1 to 3 are then applied accordingly using Eq. (A4) to derive the compliance tensor of the polycrystal $\mathbf{S}_{\mathrm{p}}$, which is then again inverted to $\mathbf{C}_{\mathrm{p}}^{\mathrm{R}}$ and indicated with $^{\mathrm{R}}$ (for Reuss).

### 2.4.2 Deriving seismic interval phase velocities for an arbitrarily anisotropic medium

The phase velocities $v_{\mathrm{ph}}(\psi, \vartheta_s)$ are obtained from the polycrystal elasticity tensor $\mathbf{C}_{\mathrm{p}}$ by applying the analytical solution to find the eigenvalues $v_{\mathrm{ph}}$ of the Christoffel matrix for an arbitrarily anisotropic medium, following Tsvankin (2001, Appendix 1A). The algorithm is presented in Appendix B and variables associated with the *cx* framework are annotated by superscript $^{\mathrm{cx}}$. Thus, the velocities are calculated for any fabric, incidence angle $\psi$, and azimuthal orientation $\vartheta_s$ of the seismic plane.

### 2.4.3 Framework comparison for cone fabrics

The frameworks (*ev* and *cx*) were compared by applying them to cone fabric for all cone angles $0° \leq \phi \leq 90°$, thus excluding any effects from asymmetric fabric. We generated artificial fabric with 1000 c-axes, randomly distributed in a solid (cone) angle, in 1° steps and calculated the respective eigenvalues. Figure 2a shows the theoretical P-wave velocity distribution $v_{\mathrm{p}}^{cx}(\psi, \phi)$ for all cone and incidence angles as calculated with the *cx* framework and Figure 2b gives the difference between the calculation results from both frameworks. As is to be expected the maximum velocity is found for a seismic wave at vertical incidence on a narrow single maximum fabric. The strongest velocity deviation between the frameworks is found for cone angles of approximately 50–60° at vertical incidence by $-1.5$ to $0.5\,\%$.

## 3 From ice core fabric to seismic velocities – case studies

We apply the *cx* framework, outlined in section 2.4, to two fabric data sets from ice cores EDML and KCC, respectively. Thus, we investigate the potential of the new framework with respect to the earlier established *ev* framework, which was already applied to fabric data from EDML (Diez et al., 2015). We use the same EDML data set (c-axis angles and grain-weighted eigenvalues), complemented by additional thin sections measured more recently. The threshold values for classifying the EDML fabric within the *ev* framework are as follows: girdle fabric is given if $\lambda_2 \geq 0.2$ and $\lambda_1 \leq 0.1$, with thick girdle fabric for $0.05 < \lambda_1 \leq 0.1$ and partial girdle for $\lambda_1 \leq 0.05$; cone fabric is identified otherwise. The threshold values for classifying the KCC fabric are chosen such that only cone fabric is recognised by the algorithm, i.e. the threshold for girdle fabric is set to $\lambda_2 \geq 0.4$ and $\lambda_1 \leq 0.1$; cone fabric is identified otherwise. This is justified by the evaluation of stereographic projections of the c-axis distributions which shows that cone fabric is dominant in all samples, although some tendencies towards girdle can be made out in deeper samples, and artificial discontinuities are prevented. KCC eigenvalues are area-weighted as grain size information is available from automatic image processing. The results obtained with the *cx* framework are considered to be more accurate for the purpose of comparing the frameworks in the following case studies. If not stated otherwise, all velocities



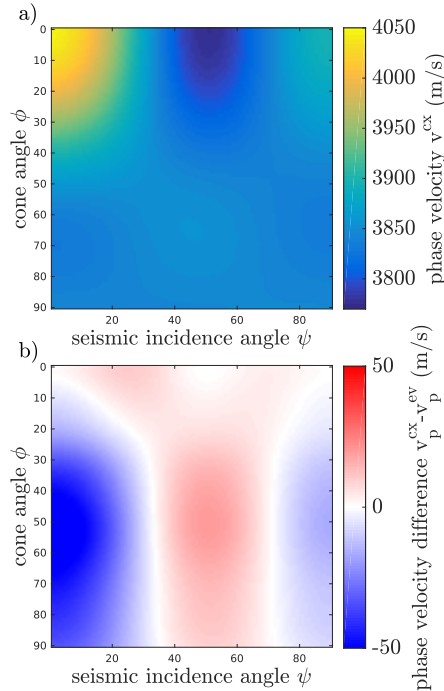

**Figure 2.** a) P-wave velocity $v_p^{cx}$ for cone fabric from randomly generated c-axes. b) Difference in P-wave velocity between the two frameworks for cone fabric. Blue color shows where the *ev* framework obtains higher velocities than the *cx* framework. Red shades indicate the opposite. They differ by $-50$ to $20\,\mathrm{m\,s^{-1}}$ (corresponding to $\leq 1.5\,\%$).

are interval velocites, i.e. the seismic wave velocity within a layer, for which an elasticity tensor is calculated based on the fabric in this layer.

### 3.1 Seismic interval velocity for vertical incidence

We assess the velocity difference between the eigenvalue and the *cx* framework $v_{p0}$ at vertical incidence of a seismic wave,

5    i.e. $\psi = 0°$ as indicated by subscript $_0$, with focus on the effect of fabric classification. Vertical incidence refers to the direction parallel to the ice core axis, which we assume to be normal to the glacier surface. As the seismic P-wave velocity for vertical incidence is invariant under azimuthal rotation of the seismic plane of the core, it is possible to assess the uncertainty introduced by using the eigenvalues. We mainly show results for the P-wave velocity, but included the S-wave velocity for the assessment of RMS velocities.

10   **Vertical incidence at EDML**

The evolution of the fabric of the EDML ice core becomes apparent from assessing the eigenvalues (Fig. 3a) and is discussed in detail in Weikusat et al. (2017). In the following, observations are made for the comparison of the velocitites from the EDML



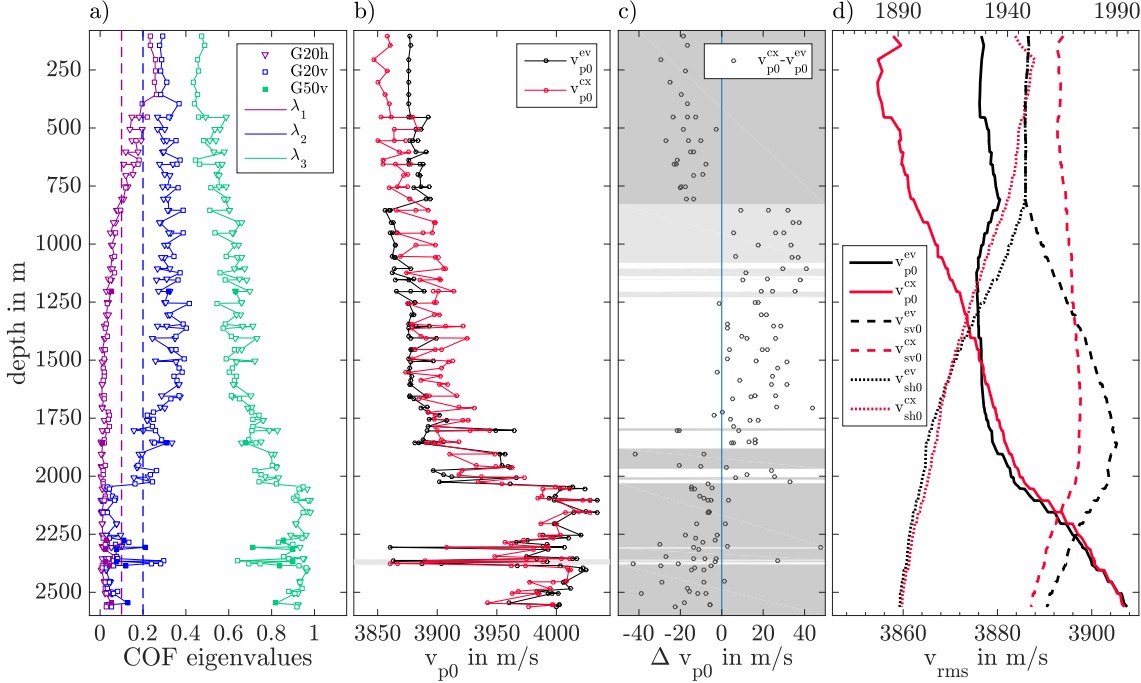

**Figure 3.** Comparison of zero-offset velocities calculated from EDML fabric data (without high resolution samples) via *ev* and *cx* framework.
**a)** Eigenvalues (symbols and solid lines for visual assistance) and threshold values (dashed lines, *ev* framework) for girdle fabric classification.
The different symbols used for eigenvalue data indicate horizontal (h, triangle) and vertical (v, square) thin sections, and the used fabric
analyser model. **b)** presents the calculated interval P-wave velocities for the two frameworks. The shaded interval around 2270 m indicates
where high resolution measurements where taken (Fig. 4). **c)** relates the difference $\Delta v_{\mathrm{p0}} = v_{\mathrm{p0}}^{\mathrm{cx}} - v_{\mathrm{p0}}^{\mathrm{ev}}$ to the fabric classes that are indicated
by shading (dark gray: cone, light gray: thick girdle, white: partial girdle). The shading extends for each data point to half the distance to
the neighbouring data points. **d)** shows the seismic RMS velocities resulting from the interval velocities integrated from the surface (without
taking density, temperature and firn into account); S-wave velocities refer to the upper x-axis and P-wave velocities to the lower x-axis.

ice core.

The general trend of the velocities of the two frameworks is in good agreement (Fig. 3b). However, a systematic difference can
be observed (Fig. 3c): for cone fabric the P-wave velocity is overestimated by the *ev* framework, for girdle fabric the P-wave
velocity is underestimated.

5 In the upper 1785 m the velocity from the *cx* framework clearly exhibits a higher variability, as quantified by the standard
deviation $s(v_{\mathrm{p0}})$ (Table 1). Below that depth, there is less variation in the velocity of the *cx* framework. The higher variability
in the eigenvalue velocity is due to several transitions between fabric classes in the depth interval of 1800 to 2035 m; notably
the prominent peak at 1802 m is clearly enhanced by this. In the lower part of the core at 2306 m a sudden weakening of the
fabric anisotropy is reflected in the results of both frameworks. The velocity is, however, underestimated by the *ev* framework
10 by 48 m s$^{-1}$ by switching from cone to girdle fabric classification. RMS velocities were calculated from the interval velocities





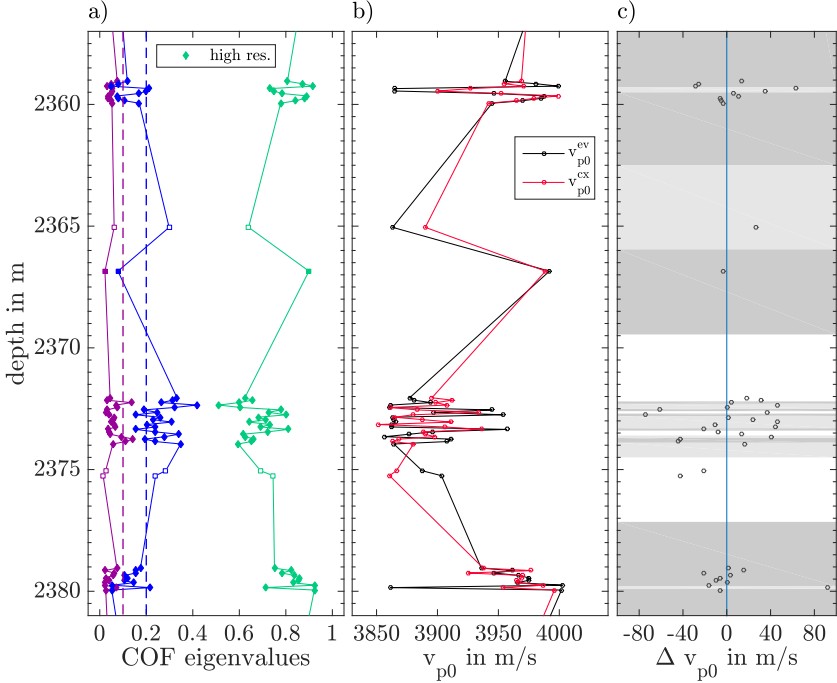

**Figure 4.** Comparison of P-wave velocities at vertical incidence, calculated from EDML fabric data measured in high resolution (vertical sections) between 2358 and 2380 m depth with the fabric analyser G50. The same variables as in Fig. 3a-c are shown: **a)** eigenvalues, with the same symbols as defined in Fig. 3 **b)** calculated interval P-wave velocity for vertical incidence **c)** the velocity difference between frameworks and fabric classes are indicated by shading (dark gray: cone, light gray: thick girdle, white: partial girdle).

for P- and S-wave (Fig. 3d) using Eq. (5) in order to assess the cumulated effect of the velocity deviation. In the anisotropic case the zero-offset RMS velocities are needed for the depth conversion in classical reflection seismic profiles (Diez et al., 2014). For EDML, the P-wave RMS velocities $v_{\mathrm{p0,rms}}$ for the two methods converge towards the bedrock as a result of the compensation of the systematic under- and overestimation described before. The S-wave RMS velocities $v_{\mathrm{s0,rms}}$ show a similar

5   shear-wave splitting (SWS) of 67 m s$^{-1}$ and 59 m s$^{-1}$ at the bedrock but the *cx* velocities also show a small split in the upper 750 m of the ice core, where the *ev* framework assumes a VTI fabric with no resulting shear-wave splitting.

Figure 4 is a close-up of the shaded depth in Fig. 3b and shows more recent high resolution measurements (filled diamonds) providing ten data points per metre. The new data exhibit a strong submetre-scale variability in fabric strength (Weikusat et al., 2017) which has not been regularly observed in ice-core fabric data so far (Fig. 4a). Only in recent ice core projects

10   fabric measurements have started to cover continuous intervals, providing new information on fabric variability. This leads to two observations: 1) Both frameworks produce fast changes in the interval velocity on the submetre scale and 2) the fabric classification of the *ev* framework switches several times within two metres. The velocities differ by up to 90 m s$^{-1}$, where the *ev* framework produces more pronounced peaks than the *cx* framework. This is the first time that the influence of strong fabric changes on the variation in seismic velocities over very short depth intervals is investigated.





**Table 1.** Standard deviation of mean interval P-wave velocities at vertical incidence for several depth intervals of the EDML ice core.

| depth in m | std. dev. $s(v_{\mathrm{p0}}^{\mathrm{ev}})$ in $\mathrm{m\,s^{-1}}$ | std. dev. $s(v_{\mathrm{p0}}^{\mathrm{cx}})$ in $\mathrm{m\,s^{-1}}$ |
|---|---|---|
| $0 - 1785$ | 10.9 | 20.3 |
| $1802 - 2035$ | 32.8 | 24.1 |
| $2045 - 2563$ | 38.5 | 36.4 |
| $2359 - 2360$ | 48.3 | 27.9 |
| $2372 - 2374$ | 32.3 | 23.0 |
| $2379 - 2380$ | 40.3 | 21.3 |

**Vertical incidence at KCC**

The fabric data is discussed in detail in a forthcoming publication (in preparation). We show the results of the velocity calculations for vertical incidence from the KCC fabric data in Fig. 5. The cone angle calculated from the eigenvalues varies between 10–30° for each depth interval (Fig. 5a). The P-wave interval velocities calculated with both frameworks (Fig. 5b) increase

with depth as a stronger anisotropic single maximum fabric evolves and show high variability between adjacent 10 cm long samples. The difference in P-wave velocity between the two frameworks is shown in Fig. 5c. The *ev* framework overestimates the P-wave velocity in average by 46 m s$^{-1}$. Hence, differences between the frameworks are similar for the KCC ice core as for the cone fabric regions of the EDML ice core. In the bottom layer the largest difference in P-wave velocity is $-135$ m s$^{-1}$, which is due to the strong single maximum that is inclined to the vertical. The change in c-axes velocity $\delta v_{\mathrm{p0}}^{\mathrm{cx}}$ of each 10 cm

sample to the previous within a continuous measurement interval can exceed 40 m s$^{-1}$ (Fig. 5d). For the estimate of P-wave and S-wave RMS velocities the average velocities for each continuous measurement interval are calculated first. Then the layer boundaries are chosen such that the measured intervals are centered within the layer as indicated in Fig. 5e by the alternating shading to obtain the RMS velocities. Neither temperature nor density corrections are applied. The difference between the framework velocities at bedrock amounts to $-39$ m s$^{-1}$ for the P-wave which corresponds to an equivalent change in estimated

depth of 1 %. The S-wave RMS velocity illustrates the shear-wave splitting which is occurring, and increasing with depth, when applying the *cx* framework and which amounts to 45 m s$^{-1}$ (2.3 %) at bedrock.

## 3.2 Seismic interval velocities for non-vertical incidence

During typical seismic profile surveys the seismic wave will have an inclined angle of incidence with respect to the vertical, normal to the glacier surface. The velocities will be changing in dependence of the incidence angle if the medium is anisotropic

and this will affect the recorded travel times (Diez and Eisen, 2015). For a single maximum (or cone) fabric that is symmetric around the vertical this angle dependency is invariant under rotation of the seismic plane. The seismic plane is the vertical



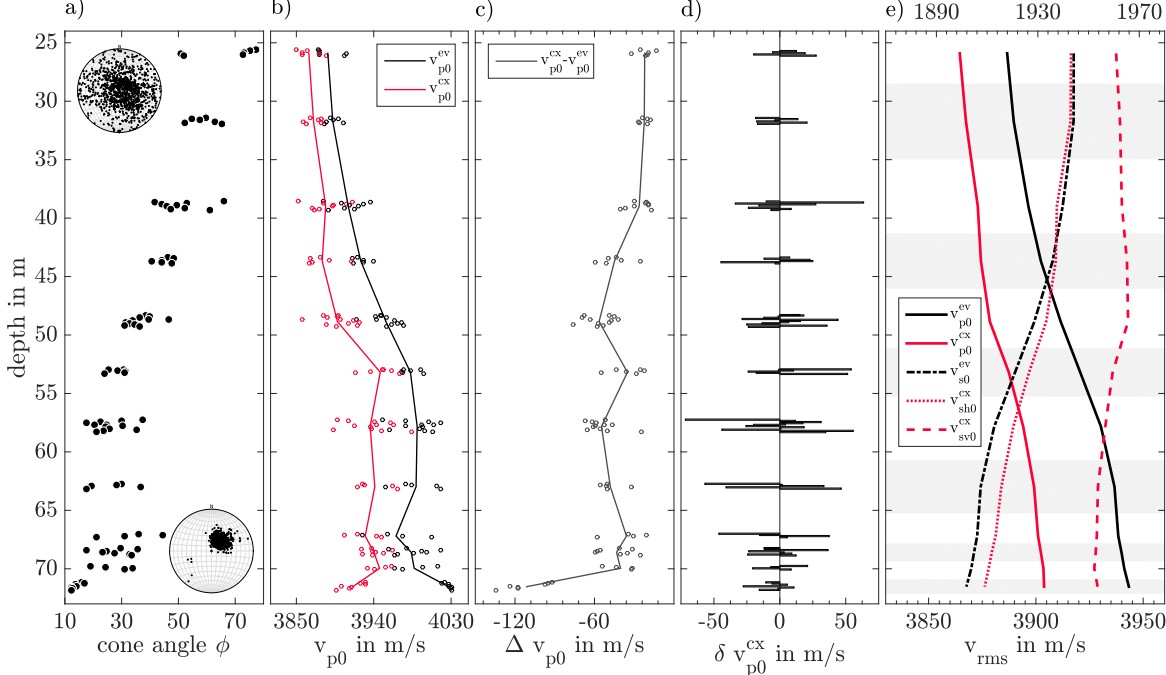

**Figure 5.** Comparison of zero-offset velocities calculated from KCC fabric data via eigenvalue and c-axis framework. **a)** shows cone angles derived from eigenvalues following Diez and Eisen (2015) and Schmidt diagrams illustrating the distribution of c-axes in the upper and the lower part of the core (projection of c-axes onto the horizontal ice core plane). **b)** presents the calculated P-wave velocities for the two frameworks for all thin sections (symbols) and the average velocities for each continuously sampled depth interval (lines). **c)** shows the difference $\Delta v_{p0} = v_{p0}^{cx} - v_{p0}^{ev}$ per sample and per interval average. **d)** illustrates the velocity change $\delta v_{p0}^{cx} = v_{p0}^{cx}(d_{i+1}) - v_{p0}^{cx}(d_i)$ between subsequent 10 cm sections at depths $d_i$. **e)** shows the RMS velocities which were calculated from the averaged velocities for layers centered around the measurement intervals; S-wave velocities refer to the upper x-axis and P-wave velocities to the lower x-axis.

$x$–$z$-plane that contains the seismic profile, which runs along the surface of the glacier in $x$-direction, and the ice core axis in $z$-direction, along which fabric information is collected (Fig. 1). However, the symmetry axis of a fabric described by a set of eigenvalues could also be inclined with respect to the vertical depending on the deformation regime on site. The recorded traveltimes will then depend on the direction of the seismic profile on the glacier surface. This, in turn, can be exploited to acquire additional information on the anisotropy. As the *cx* framework does not restrict the description of the crystal anisotropy of the effective medium to a symmetry with respect to the vertical, the variation of seismic velocities under a rotating seismic plane can be studied.

In the following we assess how the seismic velocities will change when the ice core fabric data and the seismic plane of incidence are rotated with respect to each other. The zero orientation ($\vartheta_s = 0$) is not associated with any specific geographical direction. The largest uncertainty for this assessment originates in the difficulty to identify the ice core's orientation during drilling. Although it is usually tried to fit the consecutive ice core pieces during logging to maintain the correct relative orien-





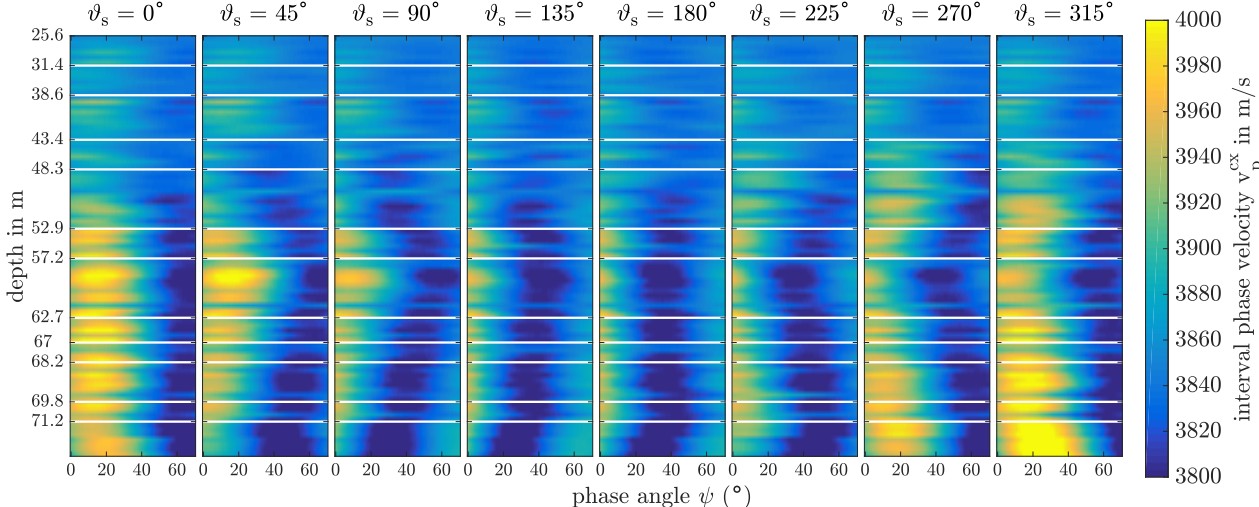

**Figure 6.** Seismic P-wave velocities for KCC calculated with *cx* framework for incidence angles up to 70° and eight seismic plane azimuth angles $\vartheta_\mathrm{s}$. The extreme values ($v_\mathrm{p0} = [3779, 4030]\,\mathrm{m\,s^{-1}}$) lie in the saturated range of the color scale for better visual contrast. Note the breaks of the depth axis (white lines), where noted depth values refer to the top of the downward extending depth interval. The thickness of the color bands is constant for each 10 cm sample.

tation it is not guaranteed that there are no discontinuities. We use the term *difference* to refer to the comparison of differently calculated velocities, while *change* is used here for the azimuth-dependent observations. In this section we focus on, and begin with, the results of the Alpine ice core KCC to demonstrate the relevance of the *cx* framework for asymmetric fabric.

**Non-vertical incidence at KCC**

The change of the P-wave velocity with increasing phase angle and rotated seismic plane as calculated with the azimuth sensitive *cx* framework is displayed in Fig. 6. The seismic plane is rotated around the ice core axis in steps of $\Delta\vartheta_\mathrm{s} = 45°$. Several core pieces were presumably rotated relative to the majority of all ice core pieces during processing to optimise the aliquot cutting. The rotation was estimated and the data is corrected accordingly before applying the *cx* framework algorithm. The influence of the asymmetry of the anisotropic fabric in the deeper part of KCC appears very clear. For some layers a spread

of velocities of up to $120\,\mathrm{m\,s^{-1}}$ is observed for a given angle of incidence when considering different seismic plane azimuth angles.

The difference between the framework velocities $v_\mathrm{p}^{cx}(\psi) - v_\mathrm{p}^{ev}(\psi)$ is shown in Fig. 7, for $\vartheta_\mathrm{s} = 0$. $v_\mathrm{p}^{ev}(\psi)$ is invariant under the rotation of the seismic plane in case of cone fabric. Thus, only the *cx* velocity is changing with rotation. The change from $v_\mathrm{p}^{cx}(\psi, \vartheta_\mathrm{s} = 0°)$ to $v_\mathrm{p}^{cx}(\psi, \vartheta_\mathrm{s})$ is shown for $\vartheta_\mathrm{s} > 0°$. The difference in P-wave velocity when comparing the calculation

frameworks reaches up to $\pm 190\,\mathrm{m\,s^{-1}}$ for the bottom layer and $\pm 50 - 100\,\mathrm{m\,s^{-1}}$ for most depths below 48 m ice depth for various incidence angles and seismic plane azimuth angles.



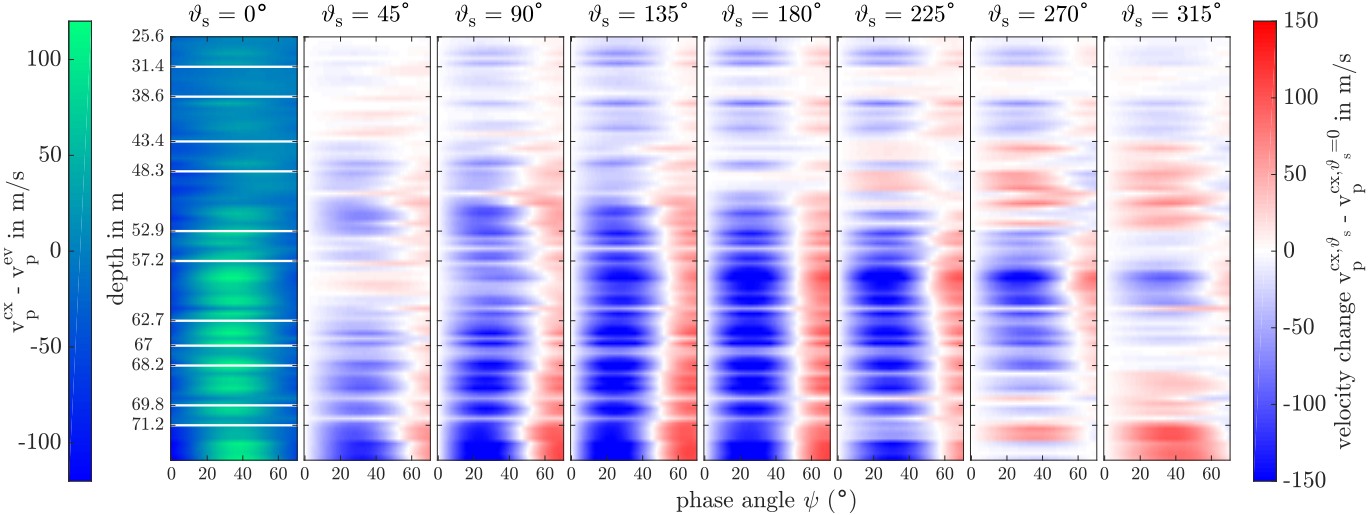

**Figure 7.** Left panel: Difference of KCC seismic P-wave velocity between *cx* and *ev* framework ($v_\mathrm{p}^\mathrm{cx} - v_\mathrm{p}^\mathrm{ev}$) for incidence angles up to 70°. The other panels show the change in P-wave velocity as calculated with *cx* framework for different seismic plane azimuth angles $\vartheta_\mathrm{s}$ compared to $\vartheta_\mathrm{s} = 0°$ ($v_\mathrm{p}^{\mathrm{cx},\vartheta_\mathrm{s}} - v_\mathrm{p}^{\mathrm{cx},\vartheta_\mathrm{s}=0}$). The extreme values ($v_\mathrm{p}^\mathrm{cx} - v_\mathrm{p}^\mathrm{ev} = [\pm 185]\,\mathrm{m\,s^{-1}}$ and $v_\mathrm{p}^{\mathrm{cx},\vartheta_\mathrm{s}} - v_\mathrm{p}^{\mathrm{cx},\vartheta_\mathrm{s}=0} = [-194, 109]\,\mathrm{m\,s^{-1}}$) lie in the saturated range of the color scale for better visual contrast, see Table 2. Note the breaks of the depth axis where noted depth values refer to the top of the downward extending depth interval.

The slower S-waves provide a better resolution and are of special interest for the study of the elastic properties of ice from traditional seismic reflection profiles (Picotti et al., 2015). In particular, the splitting of the shear waves can provide unique information about the anisotropy of the medium (Anandakrishnan et al., 1994; Smith et al., 2017). In case of the evidently asymmetric fabric of the KCC ice core we observe a shear-wave splitting of well above $200\,\mathrm{m\,s^{-1}}$ in the lower half of the ice

core with a maximum value of $281\,\mathrm{m\,s^{-1}}$. The strength of the shear-wave splitting for a particular seismic incidence angle changes when rotating the seismic plane. Figure 8 shows the difference between SV- and SH-wave velocities for non-vertical incidence (for $\vartheta_\mathrm{s} = 0°$) and investigates how the difference between the S-wave modes changes when rotating the seismic plane. The initial difference $v_\mathrm{sv}^\mathrm{cx} - v_\mathrm{sh}^\mathrm{cx}$ at $\vartheta_\mathrm{s} = 0°$ is low for small angles but for the bottom samples. It reaches more than $200\,\mathrm{m\,s^{-1}}$ for angles $> 40°$. For specific azimuth angles the change in shear-wave splitting reaches about $200\,\mathrm{m\,s^{-1}}$ for many depths below

48 m ice depth for incidence angles around $10 - 30°$. For angles above 40° the change in S-wave velocity difference reaches $-250\,\mathrm{m\,s^{-1}}$. The major part of this large change under seismic plane rotation is caused by the SV-wave variation.

**Non-vertical incidence at EDML**

No information on the core pieces' azimuth angle relative to the ice sheet or to each other is provided. However, it is assumed that no sudden short-scale change in the flow regime can occur. Thus, abrupt offsets in girdle orientation must be caused by the

unnoticed rotation of core pieces. This needs to be corrected, or at least highlighted, to avoid misinterpretation of the results from applying the *cx* framework for seismic velocity calculation considering non-vertical phase angles. For the EDML data



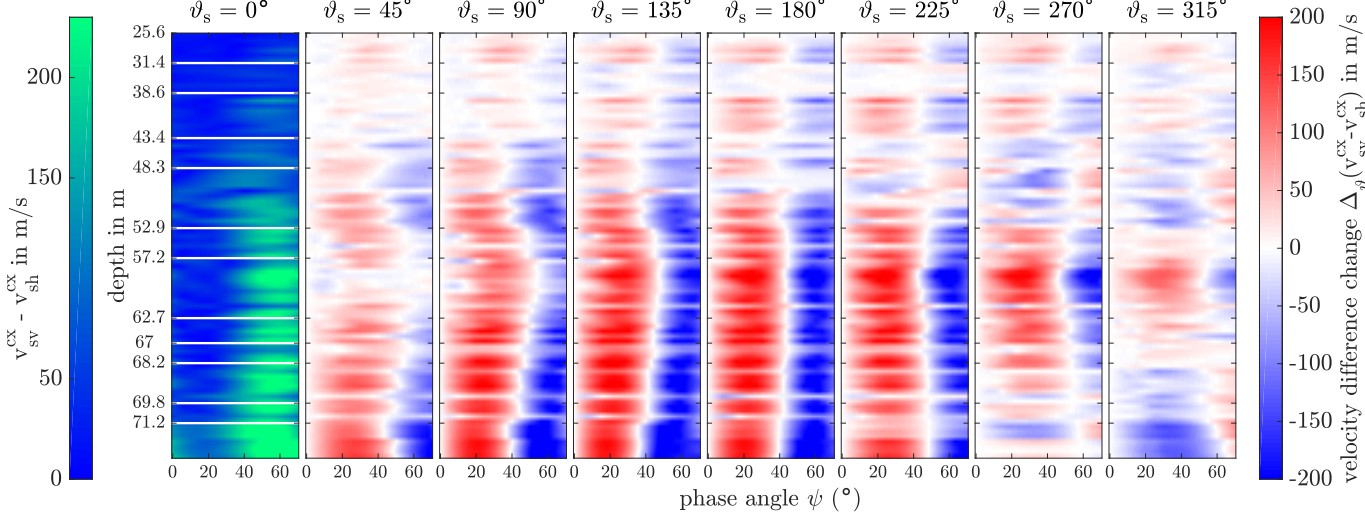

**Figure 8.** Top left: Difference of KCC seismic velocities between SH- and SV-wave as calculated with the *cx* framework for incidence angles up to 70°. The other seven panels give the change of the S-wave velocity difference for different seismic plane azimuth angles $\vartheta_s$. The extreme values lie in the saturated range of the color scale for better visual contrast, compare Table 2. Note the breaks of the depth axis where noted depth values refer to the top of the downward extending depth interval.

set the orientation of several single thin sections was corrected according to the girdle orientation of the neighbouring thin sections. A sharp change of girdle direction of about 45° in 1705 m (Weikusat et al., 2017) could not be corrected and has to be kept in mind when looking at the velocity calculation results for non-vertical incidences.

As the *ev* framework does not aim to include the orientation of the girdle, the velocity in girdle fabric is considered as invariant under the rotation as well. We only assess the change in P-wave velocity $v_p^{cx}$ as calculated with the *cx* framework. The respective figures can be found in Kerch (2016). The highest seismic P-wave velocities ($\sim 4028\,\mathrm{m\,s^{-1}}$) calculated with the *cx* framework for non-vertical incidence are found deeper than 2000 m, where the fabric anisotropy is strong, for phase angles below 20°. Seismic P-wave velocities are underestimated by the *ev* framework by max. $131\,\mathrm{m\,s^{-1}}$ and overestimated by max. $84\,\mathrm{m\,s^{-1}}$. The difference is only small ($\pm 20\,\mathrm{m\,s^{-1}}$) for cone fabric in the upper part. The highest change is apparent for the lower part of the girdle fabric, below the earlier mentioned sudden rotation of the dominant azimuth direction, and for cone fabric in the deep part of the core. There, the change in interval velocity can exceed $100\,\mathrm{m\,s^{-1}}$ for some seismic azimuth planes as compared to the defined 0°-plane.

## 4 Discussion

The velocity differences between the frameworks for the two case studies are summarised in Table 2.



**Evaluation of the *cx* framework**

The *cx* framework provides a refined approach for the use of fabric information to obtain seismic velocities in ice. By omitting the eigenvalues we keep information that is lost with the *ev* framework and we avoid the approximation to the true c-axis distribution by deriving opening angles. We average on the crystal scale to obtain the full elasticity tensor for the polycrystalline

ice. This is, to our knowledge, the first time this approach has been applied to actual ice-core fabric data. Recent work from Vaughan et al. (2017) presents P-wave velocities from cryo-EBSD data on artificial ice using the MTEX toolbox (Mainprice et al., 2011).

By using the fabric data from thin sections we acknowledge the uncertainty which arises from sampling with a relatively small sample size. We use less than 1 % of the ice core EDML and 11 % of the ice KCC to infer the fabric development in the ice

cores. There is currently no comprehensive data available to investigate the sampling effect on real ice. As we are concerned with the comparison of theoretical seismic velocities calculated from the same fabric data, we assume that the sampling uncertainty can be neglected. For the comparison with measured seismic data the uncertainty needs to be considered, as well as the appropriate density and temperature correction.

The observed variation in eigenvalues in the EDML ice core (Fig. 3a) can partly be attributed to a systematic deviation between

horizontal and vertical thin sections which is a bias produced by the older fabric analyser model G20 (Weikusat et al., 2017). Both the short-scale variation in the high resolution intervals in the EDML ice core and in the KCC ice core are not affected by the instrument bias. The *cx* framework seems to reflect this systematic variability stronger than the *ev* framework, with a higher standard deviation for the EDML depth interval $0 - 1785$ m (Table 1), illustrating the higher sensitivity to small fabric differences. The measurement of c-axes from thin sections with the instrument and the subsequent automatic image process-

ing, which provides the c-axis angles as an average per grain, contribute to a smaller extent to the overall uncertainty which is difficult to quantify. However, the processing routine (Eichler, 2013) has proven to provide robust results with respect to minor changes in the procedure and algorithm.

The currently employed algorithms for the calculation of seismic velocities in ice polycrystals on the crystal scale (including this study) do not consider any possible effects on the grain boundaries. For laboratory measurements the difference in stress

on a polycrystalline ice sample as compared with in-situ conditions can affect the degree to which grains are bonding and, thus, the elasticity (Helgerud et al., 2009). Processes like grain boundary sliding are currently explored in the context of deformation mechanism on the micro-scale (pers. comm. E.-J. Kuiper, Univ. of Utrecht, 2017) but can also influence the elastic behaviour of ice (Elvin, 1996). These issues should be addressed for future applications employing ultrasonic methods for the estimation of elastic properties of ice.

The lack of knowledge about the dispersion of seismic waves in ice introduces an unknown uncertainty to the calculation based on a monocrystal elasticity tensor that was measured in the laboratory by means of ultrasonic sounding. Again, for the application of ultrasonic methods, which operate in the same frequency range, this uncertainty can be neglected. The connection of fabric and seismic velocities on the crystal scale we present here complements this advancing field of study.

We have shown in section 2.4.3 that the *ev* and *cx* frameworks differ slightly in the case of vertically symmetric cone fabric





for vertical incidence and large cone angles. This type of fabric can commonly be expected in the shallower depth of any glacier where vertical compression is dominant. We conclude that the observed deviation in the vertical P-wave velocity profile (EDML) between the *ev* and the *cx* velocity for cone fabric could partly be attributed to this inherent difference between the frameworks.

In case of asymmetric c-axis distributions, as observed in the KCC ice core, we obtain large differences between the interval velocities of the two frameworks, resulting in a detectable difference between the RMS velocities at the bedrock which is relevant for the depth conversion. We can confirm the assessment of Voigt-Reuss bounds to lie below 1 % (for P-wave) in our study.

A main advantage of the *cx* framework is the dispensation with the fabric classification, thus eliminating artificial discontinu-

ities. In synthetic seismograms derived from the modelled velocities, such artefacts could result in artificial reflectors and, thus, lead to false interpretations. The example of high resolution sampling in the EDML ice core demonstrates the importance of this advance, allowing to separate the true high variability in seismic velocities from the artificially enhanced variability. This finding could, however, be used to tune the threshold values for the fabric classification in the *ev* framework.

**Azimuth-sensitive seismic velocities**

The *cx* framework we developed and employed in this study takes into account the asymmetry of anisotropic fabric, with respect to the vertical. This is especially relevant for glacial environments with a complex flow pattern, for example in sloping mountain glaciers, fast-flowing polar outlet glaciers (Hofstede et al., 2017, in press) and ice streams (Smith et al., 2017). For such sites the approximation of the fabric by opening angles centered around the vertical can deviate much more from the reality than for sites that are located in the vicinity of an ice divide. It becomes evident from the presented KCC case study

that the azimuthal change of the fabric and the resulting velocities are not negligible. On the contrary, the velocities calculated with the *cx* framework for non-vertical incidence angles from an arbitrary seismic azimuth can change strongly for both P- and S-wave and the associated shear-wave splitting. If the velocity depth profile changes continuously, as is illustrated in Fig. 6, 7 and 8, this should, in principle, be resolved in seismic surface profile data from different seismic azimuth directions, providing information about the (asymmetric) crystal anisotropy evolution with depth.

A requirement of the *cx* framework is the dependency on accurate core orientation information, i.e. the orientation of the fabric distribution in the equatorial plane has to be known for the consecutive fabric samples. To this date, the oriented drilling of ice cores remains a challenge. Hence, the uncertainty for the calculation of seismic velocities is much larger in the vertical direction than under azimuthal rotation. On the other hand, analysing seismic data with azimuthal resolution around an ice core drilling site could provide the information to improve the reconstruction of the core orientation.

**Rapid velocity changes over small vertical distances**

We use COF measurements on a submetre scale for our analysis of seismic velocities. The results suggest the existence of closely spaced reflective surfaces for elastic seismic waves (and also radar waves). The relevance of the presented analysis for real seismic data is based on the major assumption of a laterally extended and coherent fabric layering on the scale of



the first Fresnel zone (Drews et al., 2012). Although fabric layering is regularly observed in the KCC ice core and also in the continuously sampled depth intervals in EDML, it is still unclear how representative these short-scale variations are for both the close vicinity and a larger region in a glacier. However, evidence has been presented for abrupt COF changes as a frequent cause of seismic reflectivity (Horgan et al., 2011). Other studies do not observe such a high reflectivity due to COF

but identify a high degree of gradually evolving fabric anisotropy (Picotti et al., 2015) or single strong reflections associated with transitions in fabric classes, e.g. from cone to girdle (Diez et al., 2015). The coherence of thin layers with distinct fabric will largely depend on the unresolved question of how they evolve exactly. If the short-scale fabric stratigraphy is largely governed by local conditions and heterogenous small-scale deformation, possibly resulting in "layer roughness" (Drews et al., 2009), no coherent structure is to be expected (Diez et al., 2015). In this case, it should be challenged, how representative

the elastic properties derived from thin sections are, and the question arises, how non-coherent short-scale fabric changes alter the rheological properties of the bulk. It can be hypothesised that under the increasing influence of large-scale shear deformation in the deeper regions of the glacier coherent fabric layers might develop. Accordingly, more seismic reflectivity should be expected in depth and from more dynamic settings, as proposed by Horgan et al. (2011). Eisen et al. (2007) show that transitions in COF in the deep ice can be followed with radio-echo sounding over longer horizontal distances. However,

variations in seismic velocity on short vertical scales cannot be resolved with conventional surface-based seismic techniques with large wavelengths of the order of $10\,\mathrm{m}$, depending on the source of the seismic waves and the sounding depth. Still, Hofstede et al. (2013) obtain numerous laterally continous reflections at Halvfarryggen, Antarctica. They suggest that closely spaced layers ("stacks") of varying fabric, possibly as has been observed in this study, are the major cause for the reflections. Far more fabric data than is currently sampled in ice core studies, is required to pursue this hypothesis in the future. To

this end, ultrasonic methods can be applied in ice core boreholes (Bentley, 1972; Gusmeroli et al., 2012) to infer crystal-orientation fabric in situ. Although the interpretation of these data is not straightforward (Maurel et al., 2015), it is currently the only technique that is capable of a continuous fabric measurement. However, a sonic pulse samples the volume around the borehole ($\sim 2\,\mathrm{m}^3$, Gusmeroli et al., 2012), which means the method is not azimuth-sensitive. While it cannot provide the two-dimensional microstructure nor exact and highly resolved fabric information, it can help to bridge the gap between

laboratory-based interval fabric measurements and large-scale seismic data. Following the perceptions of the present study we recommend to include the investigation of the possible influence of variations in grain size for the seismic wave propagation in polycrystalline ice, which is currently not considered for theoretical calculations, to complement recent work on the temperature dependency of elastic properties (Vaughan et al., 2016). Ongoing microstructure studies on both Alpine and polar ice provide indications of considerable vertical short-scale variability in grain topology. Recent laser ultrasound measurements on ice have

provided first high-resolution data (Mikesell et al., 2017) and promise further advances towards understanding and efficiently measuring the elastic properties of polycrystalline ice on the crystal scale.



## 5  Conclusions and Outlook

The presented *cx* framework contributes to the understanding of the propagation of seismic velocities in glacial ice by deriving bulk elastic properties on the crystal scale. Based on anisotropic fabric from two ice cores, we showed that the fabric classification scheme in the *ev* framework can mask the true velocity variability by producing artificially enhanced peaks in the

velocity profile. By applying the *cx* framework we extract the velocity variability that is caused by the actual fabric variability. The velocity difference between the *cx* and *ev* frameworks is larger for the Alpine than for the polar core. This suggests that the *ev* framework provides a good enough approximation for the polar site, situated on an ice divide, for the current degree of seismic resolution and interpretation of physical properties, not considering the artificial discontinuities, but is not adequate for the Alpine site.

We found that the azimuthal change in P-wave velocity and shear-wave splitting can be as large as $\sim 200\,\mathrm{m\,s^{-1}}$. We conclude that the possibility of an azimuthal asymmetry of the fabric distribution should be considered when planning seismic surveys (e.g. polarimetric profiles around a drilling site) as well as for the calculation of seismic velocities from fabric data. This also offers an opportunity to further constrain azimuthal ice-core orientation.

The results of our study demonstrate that a short-scale variability in anisotropic fabric as observed in these polar and Alpine ice

cores causes a corresponding high short-scale variability in seismic interval velocities. Current laboratory fabric measurements from an ice core drilled on an ice stream also show early indications of a high fabric variability and unexpected fabric types (pers. comm. J. Eichler, 2017), offering an ideal target for extending this study to an environment with another deformation regime. Based on the presented evidence in this study the next steps should include the investigation of how a succession of short-scale fabric layers could induce englacial reflections as has been reported and hypothesised in earlier studies (Horgan

et al., 2011; Hofstede et al., 2013).

As conventional surface-based seismic surveys are not likely to resolve these short-scale variabilities, ultrasonic techniques for borehole and laboratory studies could be the solution to both issues of lost core orientation and low resolution. For this emerging field of applications, we offer further insight into what to expect from crystal-orientation fabric anisotropy in ice. Equally, our results can provide context for data collected with frozen-in seismometers in boreholes, where evidence for shear-

wave splitting on non-vertical ray paths was found (Prior et al., 2017, in review). Lastly, we want to highlight that while the depth scale of the KCC ice core differs from that of the EDML ice core by a factor of 1/35, the presented case study is another example (Eisen et al., 2003; Diez et al., 2014) of the importance of mid-latitude high-altitude glaciers as in-situ laboratories to study fundamental processes in glaciers.

*Data availability.* The eigenvalue data sets for the ice cores KCC (Kerch et al., 2016) and EDML (Weikusat et al., 2013a, b, c, d) are

published in the open-access database PANGAEA®and available upon request.





## Appendix A: Tensor transformation

A fourth-order tensor rotation is expressed as:

$$c^{\text{rot}}_{mnop} = R_{mi}R_{nj}R_{ok}R_{pl}c_{ijkl}$$

$$\text{or} \quad \mathbf{C}^{\text{rot}} = \mathbf{R} \cdot \mathbf{R} \cdot \mathbf{C} \cdot \mathbf{R}^{\top} \cdot \mathbf{R}^{\top}$$

5 The general rotation matrix in three dimensions is given by the cosines between the axes of local $\{p,q,r\}$ and global $\{x,y,z\}$ coordinate frame:

$$\mathbf{R} = \begin{pmatrix} \cos(x,p) & \cos(x,q) & \cos(x,r) \\ \cos(y,p) & \cos(y,q) & \cos(y,r) \\ \cos(z,p) & \cos(z,q) & \cos(z,r) \end{pmatrix} = \begin{pmatrix} l_1 & l_2 & l_3 \\ m_1 & m_2 & m_3 \\ n_1 & n_2 & n_3 \end{pmatrix} \tag{A1}$$

For a coordinate transformation of the monocrystal elasticity tensor $\mathbf{C}_{\text{m}}$ from crystal to global frame two basic rotations are needed, one around the $y$-axis given by the colatitude angle $\varphi$ and another around the $z$-axis with azimuth $\vartheta$:

$$10 \quad \mathbf{R}_y = \begin{pmatrix} \cos(\varphi) & 0 & \sin(\varphi) \\ 0 & 1 & 0 \\ -\sin(\varphi) & 0 & \cos(\varphi) \end{pmatrix}, \quad \mathbf{R}_z = \begin{pmatrix} \cos(\vartheta) & -\sin(\vartheta) & 0 \\ \sin(\vartheta) & \cos(\vartheta) & 0 \\ 0 & 0 & 1 \end{pmatrix} \tag{A2}$$

It is possible to express both rotations in a single rotation matrix (as done by Maurel et al., 2015, section 3).

By using Voigt notation, which mathematically implies a change of base, the rotation matrix $\mathbf{R}_{\text{C}}$ for the elasticity tensor is constructed following Sunder and Wu (1990, see appendix) using the parameterisation in Eq. (A1) and Eq. (A2) for the respective rotation:

$$\mathbf{R}_{\text{C}} =$$

$$\begin{pmatrix} l_1^2 & m_1^2 & n_1^2 & m_1 n_1 & n_1 l_1 & l_1 m_1 \\ l_2^2 & m_2^2 & n_2^2 & m_2 n_2 & n_2 l_2 & l_2 m_2 \\ l_3^2 & m_3^2 & n_3^2 & m_3 n_3 & n_3 l_3 & l_3 m_3 \\ 2l_2 l_3 & 2m_2 m_3 & 2n_2 n_3 & m_2 n_3 + m_3 n_2 & n_2 l_3 + n_3 l_2 & l_2 m_3 + l_3 m_2 \\ 2l_3 l_1 & 2m_3 m_1 & 2n_3 n_1 & m_3 n_1 + m_1 n_3 & n_3 l_1 + n_1 l_3 & l_3 m_1 + l_1 m_3 \\ 2l_1 l_2 & 2m_1 m_2 & 2n_1 n_2 & m_1 n_2 + m_2 n_1 & n_1 l_2 + n_2 l_1 & l_1 m_2 + l_2 m_1 \end{pmatrix} \tag{A3}$$

The rotation matrix $\mathbf{R}_{\text{S}}$ for the compliance tensor is given by:

$$\mathbf{R}_{\text{S}} =$$

$$\begin{pmatrix} l_1^2 & m_1^2 & n_1^2 & 2m_1 n_1 & 2n_1 l_1 & 2l_1 m_1 \\ l_2^2 & m_2^2 & n_2^2 & 2m_2 n_2 & 2n_2 l_2 & 2l_2 m_2 \\ l_3^2 & m_3^2 & n_3^2 & 2m_3 n_3 & 2n_3 l_3 & 2l_3 m_3 \\ l_2 l_3 & m_2 m_3 & n_2 n_3 & m_2 n_3 + m_3 n_2 & n_2 l_3 + n_3 l_2 & l_2 m_3 + l_3 m_2 \\ l_3 l_1 & m_3 m_1 & n_3 n_1 & m_3 n_1 + m_1 n_3 & n_3 l_1 + n_1 l_3 & l_3 m_1 + l_1 m_3 \\ l_1 l_2 & m_1 m_2 & n_1 n_2 & m_1 n_2 + m_2 n_1 & n_1 l_2 + n_2 l_1 & l_1 m_2 + l_2 m_1 \end{pmatrix} \tag{A4}$$

The expressions for $\mathbf{R}_{\text{C}}$ and $\mathbf{R}_{\text{S}}$ as given in Diez and Eisen (2015, Eq. (A.6) and (A.7)) are reversed by mistake.

## Appendix B: Analytical solution to finding eigenvalues to the elasticity tensor

20 From the characteristic polynomial of Eq. (3) a cubic equation can be derived with the substitution $\rho v^2_{\text{ph}} \to y - a/3$:

$$\det[G_{mo} - \rho v^2_{\text{ph}}\delta_{mo}] = y^3 + dy + q = 0$$





where the coefficients $d$ and $q$ follow from combinations $a$, $b$, $c$ given by the components of the Christoffel matrix $G_{mo}$:

$$a = -(G_{11} + G_{22} + G_{33})$$

$$b = G_{11}G_{22} + G_{11}G_{33} + G_{22}G_{33} - G_{12}^2 - G_{13}^2 - G_{23}^2$$

$$c = G_{11}G_{23}^2 + G_{22}G_{13}^2 + G_{33}G_{12}^2 - G_{11}G_{22}G_{33}$$

$$\quad\quad -2G_{12}G_{13}G_{23}$$

$$d = b - a^2/3$$

$$q = 2a^3/27 - ab/3 + c$$

For $k = 0, 1, 2$ the velocities $v_{\mathrm{p}}^{\mathrm{cx}}$, $v_{\mathrm{sh}}^{\mathrm{cx}}$, $v_{\mathrm{sv}}^{\mathrm{cx}}$ are found from

$$v_{\mathrm{ph}}(k) = \left\{ \left( \left( \frac{2}{\sqrt{3}}\sqrt{-d}\cos\left( \frac{1}{3}\left( \arccos\left( -\frac{q}{2\sqrt{(-d/3)^3}} \right) \right. \right. \right. \right. \right.$$
$$\left. \left. \left. \left. \left. + 2\pi k \right) \right) \right) - \frac{a}{3} \right)\rho^{-1} \right\}^{1/2} \tag{B1}$$

and are real under the conditions:

$$\frac{q^2}{4} + \frac{d^3}{27} \leq 0 \quad \text{and} \quad 0 \leq \arccos\left( -\frac{q}{2\sqrt{(-d/3)^3}} \right) \leq \pi \tag{B2}$$

The algorithm is implemented in MATLAB$^{\circledR}$ for this study.

*Author contributions.* The study was initiated and supervised by O.E. The fabric data was collected and analysed by I.W. (EDML) and J.K. (KCC), supervised by I.W. Calculations were conducted by J.K., supported by discussions with A.D. The paper was written by J.K., with comments and suggestions for improvement from all co-authors.

*Competing interests.* The authors declare that there are no competing interests.

*Acknowledgements.* We would like to thank the KCC drilling team with collegues from the Institute of Environmental Physics (Heidelberg University, Germany), the Climate Change Institute (University of Maine, USA) and the Institute for Climate and Environmental Physics (University of Bern, Switzerland). Many thanks to D.J. for suggestions for the improvement of the manuscript. J.K. was funded by the Studienstiftung des deutschen Volkes and supported by HGF grant no. VH-NG-803 to I.W.





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




**Table 2.** Summary of the results from the seismic velocity comparison between frameworks. Values are calculated depth-profile average (with standard deviation) and/or extreme ($\pm$) interval velocity differences (other than RMS). Negative values indicate smaller velocities from the *cx* framework relative to the *ev* framework. Extreme values can be influenced by outliers from the general trend. Reading example (*): For a specific seismic plane azimuth $\vartheta_\mathrm{s}$, an incidence angle $\psi$ and a specific interval at the KCC site the SV-wave velocity as calculated with the *cx* framework is found to be $279\,\mathrm{m\,s^{-1}}$ higher than is calculated with the *ev* framework which is the maximum difference for any combination of $\vartheta_\mathrm{s}, \psi$ and depth.

| Description | Notation | EDML | KCC |
|---|---|---|---|
| V-R bounds for *cx* framework | $\Delta v_\mathrm{p0}^\mathrm{cx} = v_\mathrm{p0}^\mathrm{cx} - v_\mathrm{p0}^\mathrm{cx,R}$ | $22.3 \pm 4.5\,\mathrm{m\,s^{-1}}$ | $20.9 \pm 6.0\,\mathrm{m\,s^{-1}}$ |
| Difference between framework velocities at vertical incidence | $\Delta v_\mathrm{p0} = v_\mathrm{p0}^\mathrm{cx} - v_\mathrm{p0}^\mathrm{ev}$ | $2 \pm 23\,\mathrm{m\,s^{-1}}, -74/+90\,\mathrm{m\,s^{-1}}$ | $-47 \pm 25\,\mathrm{m\,s^{-1}}$, min. $-135\,\mathrm{m\,s^{-1}}$ |
| | $\Delta v_\mathrm{sh0} = v_\mathrm{sh0}^\mathrm{cx} - v_\mathrm{sh0}^\mathrm{ev}$ | $-2 \pm 22\,\mathrm{m\,s^{-1}}, -49/+55\,\mathrm{m\,s^{-1}}$ | $9 \pm 15\,\mathrm{m\,s^{-1}}$ |
| | $\Delta v_\mathrm{sv0} = v_\mathrm{sv0}^\mathrm{cx} - v_\mathrm{sv0}^\mathrm{ev}$ | $-9 \pm 43\,\mathrm{m\,s^{-1}}, \quad -115/+110\,\mathrm{m\,s^{-1}}$ | $65 \pm 42\,\mathrm{m\,s^{-1}}$, max. $212\,\mathrm{m\,s^{-1}}$ |
| Difference between zero-offset RMS velocities at bedrock | $v_\mathrm{p0,rms}^\mathrm{cx} - v_\mathrm{p0,rms}^\mathrm{ev}$ | 0 (cancels out due to systematic bias) $-18\,\mathrm{m\,s^{-1}}$ at $750\,\mathrm{m}$ depth | $-39\,\mathrm{m\,s^{-1}}$ |
| | $v_\mathrm{sv0,rms}^\mathrm{cx} - v_\mathrm{sh0,rms}^\mathrm{cx}$ | $59\,\mathrm{m\,s^{-1}}$ | $45\,\mathrm{m\,s^{-1}}$ |
| | $v_\mathrm{sv0,rms}^\mathrm{ev} - v_\mathrm{sh0,rms}^\mathrm{ev}$ | $67\,\mathrm{m\,s^{-1}}$ | no SWS |
| Difference between framework velocities at non-vertical incidence | $v_\mathrm{p}^{cx,\vartheta_\mathrm{s}}(\psi) - v_\mathrm{p}^{ev}(\psi)$ | $-84/+131\,\mathrm{m\,s^{-1}}$ | $\pm 185\,\mathrm{m\,s^{-1}}$ |
| | $v_\mathrm{sh}^{cx,\vartheta_\mathrm{s}}(\psi) - v_\mathrm{sh}^{ev}(\psi)$ | $-184/+86\,\mathrm{m\,s^{-1}}$ | $-100/+65\,\mathrm{m\,s^{-1}}$ |
| | $v_\mathrm{sv}^{cx,\vartheta_\mathrm{s}}(\psi) - v_\mathrm{sv}^{ev}(\psi)$ | $-142/+215\,\mathrm{m\,s^{-1}}$ | $-273/+279\,\mathrm{m\,s^{-1}}$ (*) |
| Change of *cx* velocity with azimuth $\vartheta_\mathrm{s}$ | $v_\mathrm{p}^{cx,\vartheta_\mathrm{s}}(\psi) - v_\mathrm{p}^{cx,\vartheta_\mathrm{s}=0}(\psi)$ | $-97/+150\,\mathrm{m\,s^{-1}}$ | $-194/+109\,\mathrm{m\,s^{-1}}$ |
| | $v_\mathrm{sh}^{cx,\vartheta_\mathrm{s}}(\psi) - v_\mathrm{sh}^{cx,\vartheta_\mathrm{s}=0}(\psi)$ | $-73/+50\,\mathrm{m\,s^{-1}}$ | $\pm 65\,\mathrm{m\,s^{-1}}$ |
| | $v_\mathrm{sv}^{cx,\vartheta_\mathrm{s}}(\psi) - v_\mathrm{sv}^{cx,\vartheta_\mathrm{s}=0}(\psi)$ | $-210/+191\,\mathrm{m\,s^{-1}}$ | $-231/+273\,\mathrm{m\,s^{-1}}$ |
| Shear-wave splitting | $v_\mathrm{sv}^{cx,\vartheta_\mathrm{s}}(\psi) - v_\mathrm{sh}^{cx,\vartheta_\mathrm{s}}(\psi)$ | $281\,\mathrm{m\,s^{-1}}$ | $281\,\mathrm{m\,s^{-1}}$ |
| Change of shear-wave splitting with azimuth $\vartheta_\mathrm{s}$ | $\Delta_\vartheta\left(v_\mathrm{sv}^\mathrm{cx} - v_\mathrm{sh}^\mathrm{cx}\right)$ | $-177/+216\,\mathrm{m\,s^{-1}}$ | $-269/+239\,\mathrm{m\,s^{-1}}$ |
| Variability (std. dev.) of *ev* framework velocity | $s(v_\mathrm{p0}^\mathrm{ev})$ | $10$–$49\,\mathrm{m\,s^{-1}}$ (depending on depth interval) | $17\,\mathrm{m\,s^{-1}}$ (detrended) |
| Variability (std. dev.) of *cx* framework velocity | $s(v_\mathrm{p0}^\mathrm{cx})$ | $20$–$37\,\mathrm{m\,s^{-1}}$ (depending on depth interval) | $17\,\mathrm{m\,s^{-1}}$ (detrended) |
| Vertical change (between 10 cm samples) in *cx* velocity | $\delta v_\mathrm{p0}^\mathrm{cx}$ | $\sim \pm 50\,\mathrm{m\,s^{-1}}$ (high resolution intervals) | $-46/+64\,\mathrm{m\,s^{-1}}$ |
| | $\delta v_\mathrm{sh0}^\mathrm{cx}$ | | $\pm 54\,\mathrm{m\,s^{-1}}$ |
| | $\delta v_\mathrm{sv0}^\mathrm{cx}$ | | $-57/+42\,\mathrm{m\,s^{-1}}$ |