# Peer review of "Deriving micro to macro-scale seismic velocities from ice core c-axis orientations"

_The Cryosphere, 2017_

## Referee Comment (RC1) · M. Montagnat (Referee) · 5 Feb 2018

Review

"Deriving seismic velocities on the micro-scale from c-axis orientations in ice cores"

by J. Kerch et al. Submitted to The Cryosphere

Overall, the paper is very nicely written and well organised. Descriptions of the different tools, and different steps are very clear, an easy to read. I will therefore have mainly one main comment, about the purpose of the work and the way it is provided through the text.

As mentioned in the summary, the purpose of using seismic or sonic data to explore

ice anisotropy development in ice sheets and glaciers is to be able to (i) avoid using the long and small-scale technique of thin-section + analyser, (ii) to provide data including larger volume of ice and therefore more representative, and (iii) to perform more measurements, over larger areas. (There might exist other interests, but these ones are already strong). To do so, one needs to be able to invert the seismic signal into a texture data. The best would be to have access to the full orientation of every grain (c- and a-axes). In the "real life", we will mainly have access to some "symmetry" of the texture, at a polycrystalline scale, over a given volume. A symmetry that is associated with the tool used (radar...) and the inversion procedure. This is likely why previous works mentioned in the text focused on the eigenvalues of the second orientation tensor, with some symmetry hypotheses, as the likely result of this inversion. But there is no hope that this inversion could give access to the exact c-axis orientation distribution over this volume (as far as I know from the available equipment so far). Therefore it should be made very clear that the work performed in this paper is an exercise aiming at pointing the likely error deriving from the inversion procedure in some specific cases. Otherwise, making use of an existing c-axis distribution data set to obtain seismic velocities has no interest by itself. Therefore how the given algorithm (cx framework) will help to improve the inversion procedure should be made clearer in the text. For instance, could a shear-wave splitting from a seismic dataset could be directly inverted as resulting from a non-symmetric orientation distribution (cf. Abstract), and provide some information about this orientation distribution? From the comparison between the ev and cx framework performed here as an "exercise", could some specific signal be obtained to be able to assess that an experimentally obtained seismic signal is related to a non vertically clustered fabric (as in the case of the bottom of KCC)?

In the discussion part, the authors should make it clear how there work can be used to provide "safeguards" against misinterpretation of the seismic signal by a "simple" inversion toward an eigenvalue data set.

Another important point that the authors should mention is that, contrary to the c-axis

orientation angles, the eigenvalues of the second order orientation tensor do not provide a complete and unambiguous description of the texture. Indeed, one would require all the other orders of this orientation tensor to do so. By working with eigenvalues, we already work with an incomplete and bias COF data. Therefore, some variability are strongly smoothed by this procedure. Indeed, a multi-cluster type of texture will appear like a cone-angle type with the eigenvalue data, while the c-axis orientation distribution will give access to the complexity. So this is not such a result to find that the variability is better reproduced by using directly the full c-axis data...

One more point, that seems to me important but that could result from a misunderstanding from my side: The eigenvalues are provided together with the set of eigenvectors. The orientation of these eigenvectors provide the "orientation of the fabric" (if we can call it that way) relative to the thin section referential... Let's assume the thin section was done with a very strict control so that it's orientation relative to the "real" vertical is very well known, and that the core can be assumed to be vertical, then, the orientation of the eigenvector referential should tell us about some "non verticality". By making the assumption that this referential corresponds somehow to the "real" vertical, one introduce a strong assumption. This is hard to do otherwise, because of all the unknown mentioned before. BUT this assumption is not made at all when using the cx framework, since the true orientations are considered, and these orientations could be titled relative to the "real" vertical because of a tilted core, or during the thin section process... We can then expect some bias in the comparison due to this difference of consideration of this "tilt" of the fabric. This is mentioned somewhere within the text, but it does not appear to be considered as a source of differences between the velocity measurements, although is could play a strong role, especially along the KCC core. Shouldn't it be tested? For instance by aligning systematically the incidence angle with the eigenvector relative to the largest eigenvalue?

And the end, about the discussion corresponding to the layering of the core (discussion part). Couldn't this cx framework be perfectly adapted to test the effect of such a

layering on a seismic velocity data set? One could artificially vary the length of the layers, and force the anisotropy and check whether the response stands within the resolution frame of the measurements, etc. Maybe I am not qualified enough to see where the difficulty stands but it would be a nice byproduct of this cx framework?

Detailed comments:

p1. l. 20: Please site previous works done on that, or mention that Faria et al. is a review... For instance Alley 1988, Azuma 1985...

Part 2.4: maybe remind here that you are entering the cx-framework... Not so clear when reading the paper for the first time.

P7. l.1-2: here for instance, the authors refer to the error introduced in the ev framework, as if it was interesting "by itself", while it should be put back in the context of the inverse approach that aims at going from seismic data to eigenvalues (since going to real c-axis measurements will not be possible). What is the amount of signal lost? What kind of mistake could be made?? Is this inversion making sense? This is why the cx framework could be really meaningful.

P7. l. 5: maybe mention here again the fact that you call it cx framework (I was a little lost).

P8. l. 8: do you mean Appendix B (instead of 1A)?

p9. l. 4-5: here I started questioning myself about the effect of a non vertical texture that is translated into "non vertical" eigenvectors (to say it simply, see previous remarks), and that could have some impact by not being considered into the ev framework, but well integrated, per-se, into the cx framework.

P11. l. 3: how many layers do you use for the RMS calculations? How to you choose them? How does this impact the result? Same question for the case of KCC data treatment.

[Figure]

P11. l. 14: maybe put here "change in the estimated variation of seismic velocities", since these velocities are modelled and not measured... By the way, what is the resolution expected in the seismic velocity measurements? Are the differences evidenced here above or below these resolutions?

P13. l. 7: OK theoretically, but in general we don't know where is the exact vertical when analysing thin sections, and it can be tilted more than 10°, either because of inclined drilling, or because of thin section processing, or both...

p15. l. 15: about the likely misinterpretation, maybe give an illustration in the data, for clarity?

P16. l. 6: maybe put the figure in appendix at least, I was quite frustrated not to see it...

p17. l. 2: "The cx framework provides a refined approach for the use of fabric information to obtain seismic velocities in ice"... Once again, what we aim at is to obtain the fabric out of seismic data (unless there exist other purposes that should be mentioned!). So what can we learn out of this "exercise" that could help to solve the inversion problem, and to be more accurate in treating seismic data in terms of fabric information???

p17. l. 24-29: I find this paragraph highly speculative, and not necessary here... to relate stress conditions to grain bounding so quickly is speculative, and to mention the effect of GBS (that is far from being realistic along ice cores with large grain size) on elasticity is also very strong! Maybe it would be better to remain in the core of the subject?? or you would need to justify more...

p18. l. 9: Again, OK with what is said here, but since we have no hope to be able to inverse seismic data into exact c-axis orientations, how helpful is this framework?

---

## Referee Comment (RC2) · H. Horgan (Referee) · 12 Feb 2018

**Deriving seismic velocities on the micro-scale from c-axis orientations in ice cores. Kerch et al.**

February 2018

Kerch et al make a useful contribution to the literature with this study of crystal orientation fabric distributions and their resulting seismic velocities. The manuscript mainly focuses on presenting a framework by which c-axis observations can be converted to representative elastic properties, which are then used to estimate seismic phase velocities. The manuscript also spends considerable time comparing the new technique, which requires detailed knowledge of the c-axis distribution, to an already established framework that uses the more readily available eigenvalue representation of the c-axis distribution. The paper is well considered and includes findings that are useful to researchers working at both the micro and macro scales. Some clarification of the text is needed. Most of my comments below are intended to improve the presentation of the study and highlight some of the implicit and explicit findings.

To improve accessibility a flowchart type figure showing both the *ev* and *cx* framework would be a useful addition. The topic is necessarily dense, and the distinctions, while clear in the text, would be more instantly apparent in a figure. This would make it clear to the casual reader what is required as inputs, and what are the key steps that influence the result. Furthermore, to improve accessibility, Figure 1 could be thoroughly described in the introduction. This description could include details on the seismic acquisition reference plane currently described in section 3.2. Doing this would link the scales considered in the introduction.

The abstract needs some work. Mainly, it should include the key findings of the study. At present it details what will be presented but does not provide summary information of the main findings regarding the impact of azimuth, the degree of shear wave splitting, etc. Quantifying key findings in the abstract would be appropriate.

**Minor Changes**

**Title**
The title could benefit from rewording. One of the aims of this work is bridging the gap between the micro and macro scales in various ways. Your results inform both the micro scale and, through RMS velocities, the macro scale. To reflect this and to increase the audience consider something like: "Deriving micro to macro-scale seismic velocities from ice core c-axis orientations"

**Introduction**

P1 L1–2. To avoid hyperbole, I would also change the 'greatest' to 'great' and the 'urgently needed' to 'needed'. This first sentence also suffers from two objects, consider changing to 'One of the great challenges in glaciology is presented by the lack of an efficient method for estimating the bulk...' or similar.

P1. L3. 'in a glacier' to 'in ice'.

P1. L3. 'We revisit..' to 'We establish a new method of estimating seismic velocities from c-axis distributions and compare it to an already established method that uses fabric eigenvalues.'

P1. L6 'Alpine' to 'alpine' (here and elsewhere).

P1. L8 'seismic velocity as a function of horizontal azimuth' How much variation?

P1. L9 'strong azimuth-dependent shear wave splitting' How much?

P1. L10 How much change observed over what scale?

**Introduction**

P1 L15–16. '...of the ice...' to 'of ice dynamics, in which internal deformation...'

P1 L16 '...evident and described on the macro-scale..' Elaborating on this point would be useful.

P2 L7–9 '...obtain information...' to '...obtain spatially distributed information on the COF structure at various depths in the ice column, the acquisition of which would be unfeasible using direct sampling via ice-coring.'

P2 L14 '...commonly used...' to '...commonly reported...'

P2 L16 '(ev framework)' to '. (We refer to the method of Diez and Eisen (2015) as the *ev* framework.)'

P2 L17. Define EDML and EPICA on first usage.

P2 L17 'The main objective....' to 'Our main objective is to present an improved method for the estimation of the bulk elasticity tensor, and to use this to evaluate the use of the *ev* framework.'

P2 L21 'on the submetre' to 'at the submetre'

P2 L22 'effect' to 'affect'

P2 L22 'asymmetrical fabric distributions...' on what... seismic velocity.

P2 L26 'KCC' define on first usage.

**Methodology**

P2 L26 'Alpine'

P2 L26 '...European...' should be defined on first usage.

P2 L26 'until 2006' provide actual years.

P2 L29–30 '..in about 100 m distance to' '100 m from the ice core KCI' (define KCI).

P2 L31 'bed rock' 'bedrock'.

P2 L32 'minimum' 'a minimum of'.

P3 L1 'by means of' 'using'

P3 L11 'done with' 'performed with'.

P3 L12 'and covers' 'covering'.

P3 L26 'are not subject' 'Here we focus on phase velocities and group velocities are not considered.'

P4 L3 'In case' 'In the case' This sentence should state how and by whom. Currently the sentence implies that this study did this, but I don't think this is intended.

P4 L14 '"effective medium" to 'the "effective medium"'

P5 L2 '...is often used' '...is often used (cite), and we use this approach here.'.

P5 L2 'They provide' Ambiguous 'Voigt-Reuss bounds provide...'.

P5 L12 'with the density' 'where $\rho$ denotes density, $U$ denotes...'

P5 L28 'will generally not be applied' I don't think a temperature correction is ever applied here 'is not applied.'

P6 L10 'a monocrystal' 'the monocrystals'.

P6 L21, L25 'By restraining' 'By assuming'.

P6 'In fabric...' wording makes this unclear, I think you mean 'Fabric data from ice cores indicates that transitions between fabric classes usually occur gradually, and sudden change are only expected to occur due to changes in impurity content or deformation regime'.

P7 L11 'number $N_g$ of grains' 'number of grains $(N_g)$'.

p7 L11 'hundred to a thousand' is this accurate for the lowest parts of the cores.

p8 L1 'For the aim of con...' 'To determine the...'

p8 'Sm, i.e....' clarify with a new sentence 'To accomplish this, the mono...'.

P8 L4 'to $C_p^R$...' 'inverted to provide $C_p^R$, where $^R$ denotes Reuss'.

P8 L17 "by -1.5 to 0.5%" clarify what this range refers to. It's velocity differences, but state which framework is faster and what the range represents.

**From ice core fabric to seismic velocities – case studies**

P8 L22 'complemented by additional' 'complemented by XX additional...'.

P8 L22 'threshold values' 'threshold eigenvalues'.

p8 L27 'in all samples' 'in all KCC samples'.

p8 L27–28 '...girdle can be made out...' '...girdle is observed...'

p8 L29 'with the $cx$..' 'from the $cx$'.

p9 L4 'We assess' 'We now assess...'

p9 L5 'i.e. $\psi = 0°$' '(Figure 1; $\psi = 0°$)'

p9 L5 'with focus' 'with a focus'

p9 L5 'effect of fabric classification' 'affect of the $cx$ framework fabric classification'

p9 L8 'for the assessment' 'in our assessment of...'

p9 L11 'apparent from' ' apparent when'

p10 L7 'velocity is due to' 'velocity below 1785 m is due to'

p10 L8 'is clearly enhanced by this.' 'is an example of this.' Consider indicating this with an annotation on the figure.

p10 L9 'is reflected in' 'is evident in'

p10 L10 'by switching from' 'due to a switch from...'

p11 L3 'as a result of the compensation..' 'as a serendipitous result of the systematic...' As currently worded it reads like compensation is deliberate. I don't think this is the case and the results could have easily not converged.

p11 L5 '(SWS)' I don't think the acronym is used and if it is it probably is not necessary.

p11 L7 '..shows more recent high resolution measurements...' 'shows high resolution measurements completed since Diez (2015)...'

p11 L9–10 'project fabric measurements have started to cover...' '...projects have fabric measurements covered continuous intervals, providing...(cite)' Include example studies.

p11 L13 'This is the first time...' Not really the place for this kind of statement, which is more suited to the introduction or the discussion.

Just to be clear, the COF observations at depth are based on 100's–1000's of grains?

P12 L1 'The fabric data...' This sentence is not useful. Perhaps a pers comm reference is what is needed here.

P12 L7 'in average' 'on average'

p12 L9 ', which is due to...maximum that is inclined' 'due to... maximum inclined... '

p12 L15 'The S-wave...' 'The *cx* framework S-wave'

p12 L15–16 '...which is occurring and increasing with depth when applying the cx framework and which' '...Which is increasing with depth and amounts to 45...'

P12 L19 'The velocities will be changing...' 'the velocities will vary depending on the incidence angle...'

P14 L5 'phase angle' replace with 'incidence angle' (or change the incidence angles to phase angle).

p15 L1 'The slower S-waves....' Worth pointing out something like 'Although S-waves are not routinely acquired during seismic imaging in polar environments...'

p15 L3 'In case of' 'In the case of'

p15 L4 'observe a shear-wave...' 'observe S-wave...'

p15 L8 'but for the' 'except for the'

p15 L10 'around 10..' 'of 10–30...'

p15 L13 'No information....' 'For the EDML core, no information...'

p15 L15 'This needs to be...' 'Prior to the application of the *cx* framework any misorientation needs to be corrected...'

p16 L5 'We only...' 'We therefore only...'

p16 L9 'upper part' Define this depth range.

**Discussion**

P16 L14. 'The velocity...' This stand alone sentence is awkward and not a good way to begin the discussion.

P17 L1–2 'By omitting the eigenvalue.... ' 'By including all the c-axis observations, instead of using eigenvalue representation, we keep...'

P17 L12 'as well as the' 'and'

P17 L13 'temperature corrections.' 'temperature corrections are required'

P17 L17 'seems to reflect' 'propagates this systematic variability more that the *ev* framework...'

P17 L21 'which is...' 'at a level which is...'

P18 L5 'In case' 'In the case'

P18 L9 'A main advantage... is the dispensation with...' 'An advantage of the...is the lack of a need for...'

P18 L12 'allowing to' 'allowing us to'

P18 L22 'S-wave' 'S-waves'

P18 L26 'To this date' 'To date...' This is an obvious application for borehole televiewing (optical and/or acoustic.)

P18 L27 'uncertainty for the' 'uncertainty in the'

P19 L19 'non-coherent' 'incoherent'

P19 L15 'over longer horizontal' specify distance scale (e.g. 10s of km)

P19 L18 'as has' 'as have'

P19 L25 'Following the perceptions of the present study...' 'Following the findings of our study...' It would be worthwhile to include some recommendations for field acquisition seismics. For example S-waves are not routinely acquired, should they be. Are single line orientations sufficient, or single crossings adequate?

P20 L6 'Alpine'

P20 L21 'these short-scale variabilities' 'this short-scale variability'

**Figures**

Figure 1. Describe source and receiver annotations in caption. In general it would help the reader if this figure was more comprehensively described in the text.

Figure 5. It would make sense if panel a was the same as Figure 3a.

Figure 6. Phase angle and incidence angle are used interchangeably. Better to pick one.

**Tables**

Caption: 'Reading example (*):' Not sure what this refers to, clarify.

**References**

P 25 L10. This reference no longer has a clear path to publication. I suggest it is referred to as a pers comm if needed.

In closing, I thank the authors for their interesting study.

Sincerely, Huw Horgan

---

## Author Comment (AC1) · 6 Apr 2018

**Authors' response to reviews**

**Authors:** Johanna Kerch, Anja Diez, Ilka Weikusat and Olaf Eisen

**Title:** Deriving seismic velocities on the micro-scale from c-axis orientations in ice cores

Please note:

- Authors' response in **bold.**
- Our changes are in blue, also in the corresponding new manuscript (old: red).
- Line numbers mentioned here refer to the reviewed manuscript.

Response to referee #1, Maurine Montagnat

Overall, the paper is very nicely written and well organised. Descriptions of the different tools, and different steps are very clear, an easy to read. I will therefore have mainly one main comment, about the purpose of the work and the way it is provided through the text.

**We thank Maurine Montagnat for her valuable discussion and detailed comments for improving our manuscript.**

As mentioned in the summary, the purpose of using seismic or sonic data to explore ice anisotropy development in ice sheets and glaciers is to be able to (i) avoid using the long and small-scale technique of thin-section + analyser, (ii) to provide data including larger volume of ice and therefore more representative, and (iii) to perform more measurements, over larger areas. (There might exist other interests, but these ones are already strong). To do so, one needs to be able to invert the seismic signal into a texture data. The best would be to have access to the full orientation of every grain (c- and a-axes). In the "real life", we will mainly have access to some "symmetry" of the texture, at a polycrystalline scale, over a given volume. A symmetry that is associated with the tool used (radar...) and the inversion procedure. This is likely why previous works mentioned in the text focused on the eigenvalues of the second orientation tensor, with some symmetry hypotheses, as the likely result of this inversion. But there is no hope that this inversion could give access to the exact c-axis orientation distribution over this volume (as far as I know from the available equipment so far). Therefore it should be made very clear that the work performed in this paper is an exercise aiming at pointing the likely error deriving from the inversion procedure in some specific cases. Otherwise, making use of an existing c-axis distribution data set to obtain seismic velocities has no interest by itself. Therefore how the given algorithm (cx framework) will help to improve the inversion procedure should be made clearer in the text. For instance, could a shear-wave splitting from a seismic dataset could be directly inverted as resulting from a non-symmetric orientation distribution (cf. Abstract), and provide some information about this orientation distribution? From the comparison between the ev and cx framework performed here as an "exercise", could some specific signal be obtained to be able to assess that an experimentally obtained seismic signal is related to a non vertically clustered fabric (as in the case of the bottom of KCC)?

**We agree that it is very unlikely that an application of an inverse method will be able to reproduce a full crystal orientation distribution. However, our work not only aims at pointing at the error made by various approximations and is not intended for inversion purposes. Instead, we demonstrate the effect of a real fabric on seismic velocities which should be of special interest to the seismologic community. Our results may call for a reassessment of seismic velocity analysis without the inverse process being the necessary motivation, but obviously an additional opportunity to apply various frameworks.**

**Even though the presented framework is not suitable for inversion, we can use it for the**

forward calculations. To understand the variations of seismic velocities caused by variations in the crystal orientation fabric we need to be able to calculate these seismic velocities as correctly as possible. Therefore, the important objective of this study is to convey our better understanding of the distribution of seismic velocities in ice achieved so far. As we are not trying to aim for an inversion of seismic velocities to a full crystal orientation tensor we do not perform an exercise trying to derive the full crystal orientation distribution from seismic data. It is simply beyond the scope of the paper.

**We included in the introduction:**

Our main objective is to present an improved method for the estimation of the bulk elasticity tensor, and to use this to (i) evaluate the use of the *ev* framework, and (ii) demonstrate the effect of a real fabric on seismic velocities. Our study concentrates on the forward calculation of seismic velocities from the full crystal orientation distribution. The application of an inverse method will likely always require some simplification to symmetries, for which we now can quantify involved uncertainties – a required component of the covariance matrices for inverse methods.

In the discussion part, the authors should make it clear how their work can be used to provide "safeguards" against misinterpretation of the seismic signal by a "simple" inversion toward an eigenvalue data set.

**Like mentioned before, the aim of this paper is not to provide a method for inversion, but rather increase our understanding of seismic velocity distributions caused by variations in the crystal orientation distribution. Therefore, we try to increase awareness for the simplifications being made by using symmetry distributions or eigenvalues to describe seismic velocities caused by crystal anisotropy. This is the first paper where we go beyond the study of these symmetries. Setting safeguards would require a full understanding of the possible distributions and a classification of these again into some kind of symmetries or clusters, something we are avoiding by calculating velocities from the full crystal distribution. We therefore see the aim of this paper in increasing awareness and understanding of seismic anisotropy in ice rather than creating another classification scheme.**

**Nevertheless, we add a statement in the discussion of the cx framework:**

Potentially, our framework can be used in principle for the development of inverse methods to derive the fabric distribution from seismic velocities. Following experience from other fields of active seismology, this would, first, most likely require comprehensive data sets suitable for full-waveform inversion not yet available for glaciological applications; and, second, some simplifying assumptions on the distribution of crystal fabric, e.g. in terms of considered symmetries. The framework we presented allows to quantify the potential effect of simplifying assumptions and could help to more accurately specify covariance matrices, thus enabling the quantification of uncertainties coming along with the results produced by application of an inverse method.

Another important point that the authors should mention is that, contrary to the c-axis orientation angles, the eigenvalues of the second order orientation tensor do not provide a complete and unambiguous description of the texture. Indeed, one would require all the other orders of this orientation tensor to do so. By working with eigenvalues, we already work with an incomplete and bias COF data. Therefore, some variability are strongly smoothed by this procedure. Indeed, a multi-cluster type of texture will appear like a cone-angle type with the eigenvalue data, while the c-axis orientation distribution will give access to the complexity. So this is not such a result to find that the variability is better reproduced by using directly the full c-axis data...

**This is correct, the variability in seismic velocities is higher using the full crystal orientation distribution than using eigenvalues. However, this is the first study to actually use the full information to extract and demonstrate the variability. This is what we consider a noteworthy result.**

**We added to section 2.3, uncertainty of the *ev* framework:**

The eigenvalues of the second-order orientation tensor do not constitute a complete and unambiguous

description of the fabric. Specifically, they do not provide information on preferential orientations with regard to the coordinate system.

One more point, that seems to me important but that could result from a misunderstanding from my side: The eigenvalues are provided together with the set of eigenvectors. The orientation of these eigenvectors provide the "orientation of the fabric" (if we can call it that way) relative to the thin section referential... Let's assume the thin section was done with a very strict control so that it's orientation relative to the "real" vertical is very well known, and that the core can be assumed to be vertical, then, the orientation of the eigenvector referential should tell us about some "non verticality". By making the assumption that this referential corresponds somehow to the "real" vertical, one introduce a strong assumption. This is hard to do otherwise, because of all the unknown mentioned before. BUT this assumption is not made at all when using the cx framework, since the true orientations are considered, and these orientations could be titled relative to the "real" vertical because of a tilted core, or during the thin section process... We can then expect some bias in the comparison due to this difference of consideration of this "tilt" of the fabric. This is mentioned somewhere within the text, but it does not appear to be considered as a source of differences between the velocity measurements, although is could play a strong role, especially along the KCC core. Shouldn't it be tested? For instance by aligning systematically the incidence angle with the eigenvector relative to the largest eigenvalue?

**We make the assumption that the ice core axis is vertical (and discuss the effect of this assumption later) and that this axis is also the reference for the c-axis angles from the thin sections. These are the basis for calculating the eigenvalues. The ev framework works on the assumption that the eigenvector of the largest eigenvalue coincides with the vertical, without any additional information. This introduces an error when calculating seismic velocities from the ev framework. However, this is part of the errors that are introduced by the ev framework and we therefore do not want to correct for this artificially. Regardless of any tilt of the ice core axis relative to the (gravitational) vertical we compare the two frameworks within the same reference system, so there should be no bias from this. We cannot see an easy way to circumvent this problem in real application, as the question of verticality and ice-core orientation in the borehole is still not fully solved from a technical point of view.**

**For clearification, we added to the previous addition in section 2.3:**

To get a rough idea about the orientation of the fabric the eigenvectors would have to be used in addition, an approach seldomly followed. Instead, the orientation of the eigenvector to the largest eigenvalue is typically assumed to correspond to the vertical, which may in fact not be the case and could introduce an unknown uncertainty.

And the end, about the discussion corresponding to the layering of the core (discussion part). Couldn't this cx framework be perfectly adapted to test the effect of such a layering on a seismic velocity data set? One could artificially vary the length of the layers, and force the anisotropy and check whether the response stands within the resolution frame of the measurements, etc. Maybe I am not qualified enough to see where the difficulty stands but it would be a nice byproduct of this cx framework?

**This is correct, next steps following our study could and should include the modelling of synthetic seismograms to assess the effect (conclusion). However, modelling synthetic seismograms for the anisotropic case is not straight forward and would go way beyond the scope of our paper. As an example, we would like to point out to the reviewer the progress made in modelling synthetic radargrams by Eisen et al. over the last decades, which is a much simpler physical problem than the propagation of elastic waves, but still not adequately solved. The development of such a forward algorithm for elastic wave propagation is of course desirable, as it would also mark the first step to develop decent inverse methods to retrieve the fabric distribution from seismic observations.**

P1 L20: Please site previous works done on that, or mention that Faria et al. is a review.
**We noted that Faria et al. (2014) is a review paper.**

Section 2.4, P7 L5: Maybe remind here that you are entering the cx-framework.
**We added:** We refer to the new approach as *cx* framework.

P7 L1-2: the authors refer to the error introduced in the ev framework, as if it was interesting "by itself", while it should be put back in the context of the inverse approach that aims at going from seismic data to eigenvalues (since going to real c-axis measurements will not be possible). What is the amount of signal lost? What kind of mistake could be made?? Is this inversion making sense? This is why the cx framework could be really meaningful.
**We are not sure whether we understood this comment correctly, apologies if we did not. We agree that the cx framework is important and meaningful in terms of the inversion of seismic data to gain information about crystal anisotropy. However, as stated above, the scope of this paper is not the application of an inverse method to seismic data but a better understanding of the variations in seismic velocity caused by crystal anisotropy and the errors that are introduced by using simplified symmetries for the description of anisotropy. Therefore, the *ev* framework helps to increase our understanding of seismic velocity variations and is important to discuss by itself.**

P8 L8: **Appendix 1A refers to Tsvankin. As we are repeating it in Appendix B of our paper we removed the confusing reference.**

P9 L4-5: here I started questioning myself about the effect of a non vertical texture that is translated into "non vertical" eigenvectors (to say it simply, see previous remarks), and that could have some impact by not being considered into the ev framework, but well integrated, per-se, into the cx framework.
**See above explanation.**

P11 L3: how many layers do you use for the RMS calculations? How to you choose them? How does this impact the result? Same question for the case of KCC data treatment.
**For EDML we use all available data for the calculation without a selection and center the layers around the data points. We have no information on which layers could be more dominating, which is why we do not make any additional assumptions. The average layer thickness is 16 m. For KCC we first calculate the average speed in each continuous measurement interval (as explained in the text) and then extend the layers to half the distance to the neighboring intervals (12 layers).**

P11 L14: maybe put here "change in the estimated variation of seismic velocities", since these velocities are modelled and not measured... By the way, what is the resolution expected in the seismic velocity measurements? Are the differences evidenced here above or below these resolutions?
**The sentence you are referring to was deleted on the 2ⁿᵈ reviewer's request. The resolution of conventional surface based seismic surveys is lower than the fabric-based seismic velocity variations (see conclusion).**

P13 L7: OK theoretically, but in general we don't know where is the exact vertical when analysing thin sections, and it can be tilted more than 10°, either because of inclined drilling, or because of thin section processing, or both...
**You are of course right that a drilling inclination would mean that ice core axis and vertical diverge. Ideally, a known borehole inclination would be part of our framework (and any thin section analysis), but then there is still the problem of the ice core orientation, which would**

also be needed for a correction. (We think we can safely neglect the thin section processing as contribution to this uncertainty as it is a standardised procedure starting from the sawing of the ice core.) Therefore, without further information, we make the assumption that ice core axis and vertical coincide (section 2). A possible inclination does not eliminate the ability of our cx framework to resolve the azimuthal variation of a non-symmetrical c-axis distribution. However, we did not mention this additional source of uncertainty before and added in the discussion of azimuth-sensitive seismic velocities:

The appearance of a non-symmetric fabric might also be induced by inclined drilling. Ideally, to be able to link calculated and measured seismic velocities a possible inclination of the ice core with respect to the vertical and to the horizontal seismic profile should be considered.

P15 L15: about the likely misinterpretation, maybe give an illustration in the data, for clarity?

A new figure (see next comment) illustrates that in several depths the velocity in dependence of the incidence angle appears shifted from the neighboring samples. This could be caused by the misorientation of neighbouring ice core pieces or it could be a true variation. We wanted to alert the reader to be aware of this issue. The specific depth, where the core orientation appears lost (1705 m), is based on the Schmidt diagrams (not shown) but this could be the case for more depths. We included a remark in the caption of the new figure pointing it out.

P16 L6: maybe put the figure in appendix at least, I was quite frustrated not to see it...

Sorry for that, we did not want to cause any unnecessary frustration but did not include the figure in the beginning for length constraints. However, as per your request, we now included the figure in section 3.2 (Fig. 10) showing P-wave velocity difference between the frameworks and azimuthal change for EDML.

P17 L2: "The cx framework provides a refined approach for the use of fabric information to obtain seismic velocities in ice"... Once again, what we aim at is to obtain the fabric out of seismic data (unless there exist other purposes that should be mentioned!). So what can we learn out of this "exercise" that could help to solve the inversion problem, and to be more accurate in treating seismic data in terms of fabric information???

Thanks for your comment. As mentioned before the aim of the ev framework is not to apply an inverse method to seismic data. We know that one of the main goals will be to derive reliable information about crystal anisotropy from seismic data and this will only be possible using simplified symmetries, however, to be able to do so reliably we have to understand the system and possible errors. Therefore, the cx framework is in itself important to gain reliable and most correct information for the forward calculation of seismic velocities in anisotropic ice.

P17 L24-29: I find this paragraph highly speculative, and not necessary here... to relate stress conditions to grain bounding so quickly is speculative, and to mention the effect of GBS (that is far from being realistic along ice cores with large grain size) on elasticity is also very strong! Maybe it would be better to remain in the core of the subject?? or you would need to justify more...

We agree that this paragraph is not mandatory. However, for completeness, we want to address additional processes on the crystal scale that might play a role, especially when measuring in the laboratory. More and more studies focus on the measurement of ultrasonic velocities on ice cores to derive information of crystal orientation on the microscale in high resolution with the aim of finding better connection to the macroscale observations. Therefore, we would like to keep this paragraph with the cited papers giving the necessary context.

P18 L9: Again, OK with what is said here, but since we have no hope to be able to inverse seismic data into exact c-axis orientations, how helpful is this framework?

**Like mentioned before, the study of seismic velocities in anisotropic ice should not be reduced to the only goal of an inversion scheme. Setting up a well working and powerful inversion of seismic velocities or even the full wave form requires a deeper understanding of the processes and possible variations. If we only always discuss and analyse highly simplified versions we cannot gain any more in-depth understanding of the system. In fact, we could, but to quantify the errors coming along with that we first need to understand the full approach. Therefore, if we are able to derive more accurate seismic velocities from anisotropic ice we should do so. Then, in a next step, we can simplify again such a system again to (maybe) be able to invert data, with the additional knowledge we gained from the more precise forward calculations.**

Response to referee #2, Huw Horgan

Kerch et al make a useful contribution to the literature with this study of crystal orientation fabric distributions and their resulting seismic velocities. The manuscript mainly focuses on presenting a framework by which c-axis observations can be converted to representative elastic properties, which are then used to estimate seismic phase velocities. The manuscript also spends considerable time comparing the new technique, which requires detailed knowledge of the c-axis distribution, to an already established framework that uses the more readily available eigenvalue representation of the c-axis distribution. The paper is well considered and includes findings that are useful to researchers working at both the micro and macro scales. Some clarification of the text is needed. Most of my comments below are intended to improve the presentation of the study and highlight some of the implicit and explicit findings.

**We thank Huw Horgan for his appreciation and detailed comments on language and content for improving our manuscript.**

To improve accessibility a flowchart type figure showing both the ev and cx framework would be a useful addition. The topic is necessarily dense, and the distinctions, while clear in the text, would be more instantly apparent in a figure. This would make it clear to the casual reader what is required as inputs, and what are the key steps that influence the result.

**We prepared a flowchart type figure (new figure 2 in the revised manuscript) and added it to section 2.4. We added to section 2.4:**

Both frameworks are summarised in Figure 2.

Furthermore, to improve accessibility, Figure 1 could be thoroughly described in the introduction. This description could include details on the seismic acquisition reference plane currently described in section 3.2. Doing this would link the scales considered in the introduction.

**We moved the explanation of the seismic plane from section 3.2 to the introduction. The entire paragraph now reads:**

Currently, the development and extent of fabric anisotropy is mainly investigated by laboratory measurements on ice core samples which provide one-dimensional data (along the core axis, $z$-axis in Fig. 1) that only partially cover the length of the core. However, geophysical evidence of crystal-orientation fabric can also be obtained by exploiting the elastic anisotropy which influences the propagation of seismic waves in ice (Blankenship et al., 1987; Smith et al., 2017). Seismic waves propagate between a seismic source and the seismic receivers on the glacier surface within the seismic plane  (Fig. 1). This is the vertical plane underneath the horizontal seismic profile, which runs along the surface of the glacier, and may also contain the vertical ice core axis, along which fabric information is collected.

Seismic reflections occur due to sudden changes of fabric (Horgan et al., 2008, 2011; Hofstede et al., 2013) and offer the chance to obtain spatially distributed information on the COF structure in various depths of the ice column, the acquisition of which would be unfeasible using direct sampling via ice-coring.

The abstract needs some work. Mainly, it should include the key findings of the study. At present it details what will be presented but does not provide summary information of the main findings regarding the impact of azimuth, the degree of shear wave splitting, etc. Quantifying key findings in the abstract would be appropriate.

**We changed the abstract to provide summary information of the main findings. Comments on P1 L1-10 are included in the rewritten abstract. Please note the rewritten abstract in the revised manuscript.**

The title could benefit from rewording. One of the aims of this work is bridging the gap between the micro and macro scales in various ways. Your results inform both the micro scale and, through RMS velocities, the macro scale. To reflect this and to increase the audience consider something like: "Deriving micro to macro-scale seismic velocities from ice core c-axis orientations"

**We like the new title suggestion:**

Deriving micro to macro-scale seismic velocities from ice core c-axis orientations

**All comments referring to a specific page and line which are not detailed below were changed in the manuscript as suggested.**

*Introduction*

P1 L16: Elaborating on this point would be useful.
**Changed to:** ...evident and described on a macro-scale as most observations rely on remote sensing or ice sheet modeling.

P2 L17: **changed to:** ...the polar ice core EDML (European Project for Ice Coring in Antarctica in Dronning Maud Land).

P2 L22: 'effect' to 'affect'
**Not changed, we think 'effect' is the correct term here:**
Finally, we assess the effect of asymmetrical fabric distributions...

P2 L26: 'KCC' define on first usage.
**Not changed: KCC is simply the name for this ice core to fit into a pattern (KCI further down the flowline and CC ("Climate/Chemical Core") on the same altitude but on a flowline on the other side of the saddle). We think that there is no benefit for the reader to have this explained.**

*Methodology*

P4 L3: This sentence should state how and by whom.
**Changed to:** ...the components of the elasticity tensor were measured in the laboratory by means of Brillouin spectroscopy (Gammon et al., 1983).

P5 L2: '...is often used (cite), and we use this approach here.'
**Changed to:** ... is often used (Nanthikesan and Sunder, 1994; Bons and Cox, 1994; Helgerud et al., 2009; Vaughan et al., 2016), and we also use this approach here.

P5 L12: 'with the density ρ' 'where ρ denotes density, U denotes...'
**Not changed: We would prefer to keep the definitions as short as possible.**

P7 L11: 'hundred to a thousand' is this accurate for the lowest parts of the cores?
**The number of grains is highly variable throughout the KCC core with a minimum number of 155 in medium parts of the core, a maximum of 1707 grains and more than 250 grains per section in the lowest 5 m. For EDML the number of grains is between 48 and 648 with 27 samples with less than 100 grains, 24 of which are from the depth interval 2300-2380 m, but not in the deepest part.**
**We added a similar sentence in the manuscript:**
Specifically, for EDML the number of grains is mostly between 100 and 650 grains per sample with the exception of some large-grained samples from the depth interval $2300 - 2380$ m with less than 100 grains. For KCC the number of grains is between 155 and 1707 grains per sample and there are more than 250 grains in the lowest 5 m of the ice core.

P8 L17: "by -1.5 to 0.5%" clarify what this range refers to. It's velocity differences, but state which framework is faster and what the range represents.
**Changed to:** The strongest velocity deviation between the frameworks is found for cone angles of approximately 50-60° at vertical incidence, where the *ev* velocity exceeds the *cx* velocity by approx. 50 m/s (absolute deviation of 1.5 %).

*From ice core fabric to seismic velocities – case studies*

P10 L8: Consider indicating this with an annotation on the figure.
**We included an additional depth annotation in Figure 4.**

P11 L9-10: Include example studies.
**Changed to:** Only in recent ice core projects have fabric measurements covered continuous intervals (ongoing measurements at the site of the East Greenland Ice-Core Project (EGRIP); North Greenland Eemian Ice Drilling (NEEM), Eichler et al., 2013), providing new information on fabric variability.

P11 L13: Not really the place for this kind of statement, which is more suited to the introduction or the discussion.
**We removed the statement as we have a similar statement already in both abstract and conclusion.**
Just to be clear, the COF observations at depth are based on 100's–1000's of grains?
**See explanation for P7 L11.**

P12 L1: 'The fabric data is discussed in detail in a forthcoming publication (in preparation).' This sentence is not useful. Perhaps a pers comm reference is what is needed here.
**We changed the paragraph to:**
We show the results of the velocity calculation for vertical incidence from the KCC fabric data in Fig. 6. The cone angle calculated from the eigenvalues varies between 10-30° for each depth interval (Fig. 6a). A detailed discussion of the fabric data is beyond the scope of this paper.

P16 L9: Define this depth range.
**We added:** ...upper part (0-800 m).

*Discussion*

P16 L14: This stand alone sentence is awkward and not a good way to begin the discussion.
**We moved the sentence to the end of the first paragraph of section 3 (From ice core fabric to seismic velocities – case studies).**

P18 L26: This is an obvious application for borehole televiewing (optical and/or acoustic.)
**We assume you refer to e.g. Hubbard et al. (2008) who discussed "digital optical televiewing of ice boreholes". To our knowledge this has not been employed to deep boreholes (below 1500 or 2000 m, with problems below because of sealing tightness, although deeper tests at NEEM were tried) and there are some issues with the available instruments regarding the operating temperature in polar environments. We chose not to include a discussion of possible solutions to the problem of oriented vertical drilling other than the implications of our results.**

P19 L15: 'over longer horizontal' specify distance scale (e.g. 10s of km)
**Changed to:** ...over longer horizontal distances of several kilometres.

P19 L25: It would be worthwhile to include some recommendations for field acquisition seismics. For example S-waves are not routinely acquired, should they be. Are single line orientations sufficient, or single crossings adequate?
**Changed to:** Following the findings of our study we recommend for seismic data acquisition in the field to (1) consider polarimetric survey setups (with two or even more crosslines) with both reflection and wideangle measurements, and to (2) focus on accurate traveltime recordings at high source frequencies. This should be supported by 3-component vertical seismic profiling where boreholes are available. Also, S-waves should be acquired as they provide useful information on crystal anisotropy due to shear-wave splitting.
On the crystal scale, we suggest to include the investigation...

*Figures*

Figure 1: Describe source and receiver annotations in caption.
**To the caption we added:**
The star symbolises the seismic source and the triangles represent a line of seismic receivers.
In general it would help the reader if this figure was more comprehensively described in the text.
**In addition to the description of the seismic plane in the introduction we added in section 2.1 (this is partially repeated in section 2.2):**
Figure 1 illustrates the geometric relation between the angles for describing the c-axis orientations from an ice core and the setup of a seismic survey profile across an ice core. The incidence of a seismic wave on an ice sample is determined by the angle of incidence $\psi$ and the azimuth angle $\theta_s$ of the seismic plane.
**We also simplified the layout of Figure 1 without changing the content.**

Figure 5: It would make sense if panel a was the same as Figure 3a.
**We agree with your suggestion. However, the eigenvalue data for KCC is not published yet (paper in preparation), which is why we would prefer not to already show the data here.**

Figure 6: Phase angle and incidence angle are used interchangeably. Better to pick one.
**We picked** angle of incidence **for figures 6, 7 and 8 and also changed the term in the text (3 x).**

*Tables*

'Reading example (*):' Not sure what this refers to, clarify.

**The asterisk marks one line in the table (we moved it to the front of the line) to give an example as to how to read the table. To distinguish this further from the table caption we moved the example to be a table footnote.**

**We also clarified in the table caption that all extreme values are given for incidence values of 0–70° and corrected the extreme values for S-wave difference between the frameworks accordingly (before: extreme values for $0 - 90°$, where the extreme values were found for angles $> 70°$).**

*References*

P25 L20: This reference no longer has a clear path to publication. I suggest it is referred to as a pers comm if needed.

**We agreed on a pers. comm. reference with D. Prior.**

---

## Author Response (AR2)

*Author's response to Editor Decision*

**Deriving micro to macro-scale seismic velocities from ice core c-axis orientations**

J. Kerch et al.

Comment by editor (Martin Schneebeli, 21 April 2018):

I trust you that you incorporate the important comments made by Dr. Montagnat in the final version. So I do not request a further minor revision, only a technical correction.

Response:

We changed "full crystal orientation" to "c-axis orientation" on p3, l3 and included Maurel et al. (2015) on p2, l27 as requested by Dr. Montagnat.
Additionally, we added another data set used in the study to the data availability declaration which is now published in the open-access database PANGAEA (Weikusat et al., 2018).

[revised manuscript text omitted]